# Metabolomics identifies and validates serum androstenedione as novel biomarker for diagnosing primary angle closure glaucoma and predicting the visual field progression

Shengjie Li[1,2,3,4,5]*[†], Jun Ren[1][†], Zhendong Jiang[1][†], Yichao Qiu[1], Mingxi Shao[1], Yingzhu Li[1], Jianing Wu[1], Yunxiao Song[6], Xinghuai Sun[2,3,4,5], Shunxiang Gao[7,8]*, Wenjun Cao[1,2,3,4,5]*

[1]Department of Clinical Laboratory, Eye & ENT Hospital, Shanghai Medical College, Fudan University, Shanghai, China; [2]Department of Ophthalmology & Visual Science, Eye & ENT Hospital, Shanghai Medical College, Fudan University, Shanghai, China; [3]State Key Laboratory of Medical Neurobiology, Institutes of Brain Science, Fudan University, Shanghai, China; [4]Key Laboratory of Myopia, Chinese Academy of Medical Sciences, Shanghai, China; [5]NHC Key Laboratory of Myopia, Fudan University, Shanghai, China; [6]Department of Clinical Laboratory, Shanghai Xuhui Central Hospital, Fudan University, Shanghai, China; [7]Department of Ophthalmology, Shanghai General Hospital, Shanghai Jiao Tong University School of Medicine, Shanghai, China; [8]National Clinical Research Center for Eye Diseases, Shanghai Key Laboratory of Ocular Fundus Diseases, Shanghai Engineering Center for Visual Science and Photomedicine, Shanghai, China

*For correspondence:
lishengjie6363020@163.com (SL);
shunxianggao@163.com (SG);
wgkjyk@aliyun.com (WC)

[†]These authors contributed equally to this work

## Abstract

**Background:** Primary angle closure glaucoma (PACG) is the leading cause of irreversible blindness in Asia, and no reliable, effective diagnostic, and predictive biomarkers are used in clinical routines. A growing body of evidence shows metabolic alterations in patients with glaucoma. We aimed to develop and validate potential metabolite biomarkers to diagnose and predict the visual field progression of PACG.

**Methods:** Here, we used a five-phase (discovery phase, validation phase 1, validation phase 2, supplementary phase, and cohort phase) multicenter (EENT hospital, Shanghai Xuhui Central Hospital), cross-sectional, prospective cohort study designed to perform widely targeted metabolomics and chemiluminescence immunoassay to determine candidate biomarkers. Five machine learning (random forest, support vector machine, lasso, K-nearest neighbor, and GaussianNaive Bayes [NB]) approaches were used to identify an optimal algorithm. The discrimination ability was evaluated using the area under the receiver operating characteristic curve (AUC). Calibration was assessed by Hosmer-Lemeshow tests and calibration plots.

**Results:** Studied serum samples were collected from 616 participants, and 1464 metabolites were identified. Machine learning algorithm determines that androstenedione exhibited excellent discrimination and acceptable calibration in discriminating PACG across the discovery phase (discovery set 1, AUCs=1.0 [95% CI, 1.00–1.00]; discovery set 2, AUCs = 0.85 [95% CI, 0.80–0.90]) and validation phases (internal validation, AUCs = 0.86 [95% CI, 0.81–0.91]; external validation, AUCs = 0.87 [95% CI, 0.80–0.95]). Androstenedione also exhibited a higher AUC (0.92–0.98) to discriminate the

severity of PACG. In the supplemental phase, serum androstenedione levels were consistent with those in aqueous humor (r=0.82, p=0.038) and significantly (p=0.021) decreased after treatment. Further, cohort phase demonstrates that higher baseline androstenedione levels (hazard ratio = 2.71 [95% CI: 1.199–6.104], p=0.017) were associated with faster visual field progression.

**Conclusions:** Our study identifies serum androstenedione as a potential biomarker for diagnosing PACG and indicating visual field progression.

**Funding:** This work was supported by Youth Medical Talents – Clinical Laboratory Practitioner Program (2022-65), the National Natural Science Foundation of China (82302582), Shanghai Municipal Health Commission Project (20224Y0317), and Higher Education Industry-Academic-Research Innovation Fund of China (2023JQ006).

## eLife assessment

This study presents a **valuable** finding that serum androstenedione levels may provide a new biomarker for early detection and progression of glaucoma, although a single biomarker is unlikely to be singularly predictive due to the etiological heterogeneity of the disease. The strength of the evidence presented is **solid**, supported by multiple lines of evidence.

## Introduction

Glaucoma is the most frequent cause of irreversible blindness worldwide (*Jonas et al., 2017*), and its prevalence is increasing globally, making it a public health concern (*Liu et al., 2023*). The primary form of glaucoma worldwide is primary open angle glaucoma (POAG). However, in East Asian populations, primary angle closure glaucoma (PACG) is predominant, affecting 70% of glaucoma patients globally (*Song et al., 2011*; *Stein et al., 2011*; *Tham et al., 2014*). Hence, early detection of PACG and accurate prediction of visual field (VF) changes can potentially preserve vision and mitigate the risk of PACG advancement. Measurement of intraocular pressure (IOP), perimetry, gonioscopy, and optical coherence tomography is the main diagnostic testing to assess for glaucoma and to monitor for disease progression (*Stein et al., 2021*). However, the present clinical techniques are inadequate for the early diagnosis and prognosis of VF progression as they depend on specialized eye examination equipment and the expertise of ophthalmologists. Furthermore, patients seldom seek the services of an ophthalmologist until the symptoms worsen or visual acuity significantly deteriorates. Importantly, the *He et al., 2019*, study taught us that performing laser iridotomy on patients with 180 degrees of angle closure does not prevent PACG, thus gonioscopy may not be a good tool to help distinguish PACG from controls. Therefore, the development of a straightforward and dependable biomarker or biomarker panel to facilitate early detection and prognostication of progressive VF loss in PACG is imperative, rather than relying solely on the expertise of ophthalmologists and specialized equipment.

Multiple etiology and risk factors lead to the onset/development of PACG in humans, which involves a tremendous flow of physiological changes, genetic factors, and metabolic adaptations (*Sun et al., 2017*). Both physiological changes, genetic factors, and metabolic adaptations should lead to profound changes in most metabolic pathways. Small molecule metabolites, as crucial biomarkers of cellular function (*Yan et al., 2022*), which identifies and quantified by metabolomics, are the omics product closest to clinical phenotypes (*Guijas et al., 2018*). A burgeoning corpus of evidence indicates metabolic changes in individuals afflicted with glaucoma (*Hysi et al., 2019*; *Kang et al., 2022*; *Rong et al., 2017*; *Tang et al., 2021*; *Wang et al., 2021*; *Qin et al., 2022*). Nevertheless, extant metabolomics research on glaucoma is constrained by inadequate sample sizes, a dearth of validation sets to corroborate findings, and an absence of specificity analyses. Notably, investigations have yet to comprehensively characterize the serum metabolome in sizable cohorts to identify putative biomarkers capable of distinguishing patients with PACG from healthy controls.

In this study, a cross-sectional and prospective cohort design was employed to systematically profile blood metabolites using widely targeted metabolomics and chemiluminescence immunoassay in both patients with PACG and control individuals. The objectives of the study were to characterize the metabolic profile associated with PACG, identify potential blood diagnostic biomarkers of PACG, assess the specificity of diagnostic biomarkers for PACG of any severity, and verify the biomarkers used to predict the VF progression of PACG.

## Methods

### Participant

From January 2020 to December 2021, newly diagnosed PACG and age-sex-matched controls were recruited from the Eye Center of Fudan University and Shanghai Xuhui Central Hospital. Detailed ophthalmic examinations and medical examinations were described in the supplementary material. Approval from the Institutional Review Board/Ethics Committee (2020[2020013]) was obtained from the Ethics Committee of the Eye and ENT Hospital, and the study adhered to the principles of the Declaration of Helsinki. Informed consent was obtained from all subjects.

A glaucoma specialist diagnosed PACG: The diagnostic criteria, inclusion, and exclusion criteria for PACG were described previously (*Li et al., 2023*; *Li et al., 2021*; *Li et al., 2020*). The methods performed for VF analysis were done as previously described (*Li et al., 2023*; *Li et al., 2020*). Previously described methods (*Chen et al., 2018*; *Li et al., 2023*; *Li et al., 2020*; *Naghizadeh and Holló, 2014*) were performed for the determination of functional PACG VF loss progression according to an event-based analysis modified for Octopus perimetry.

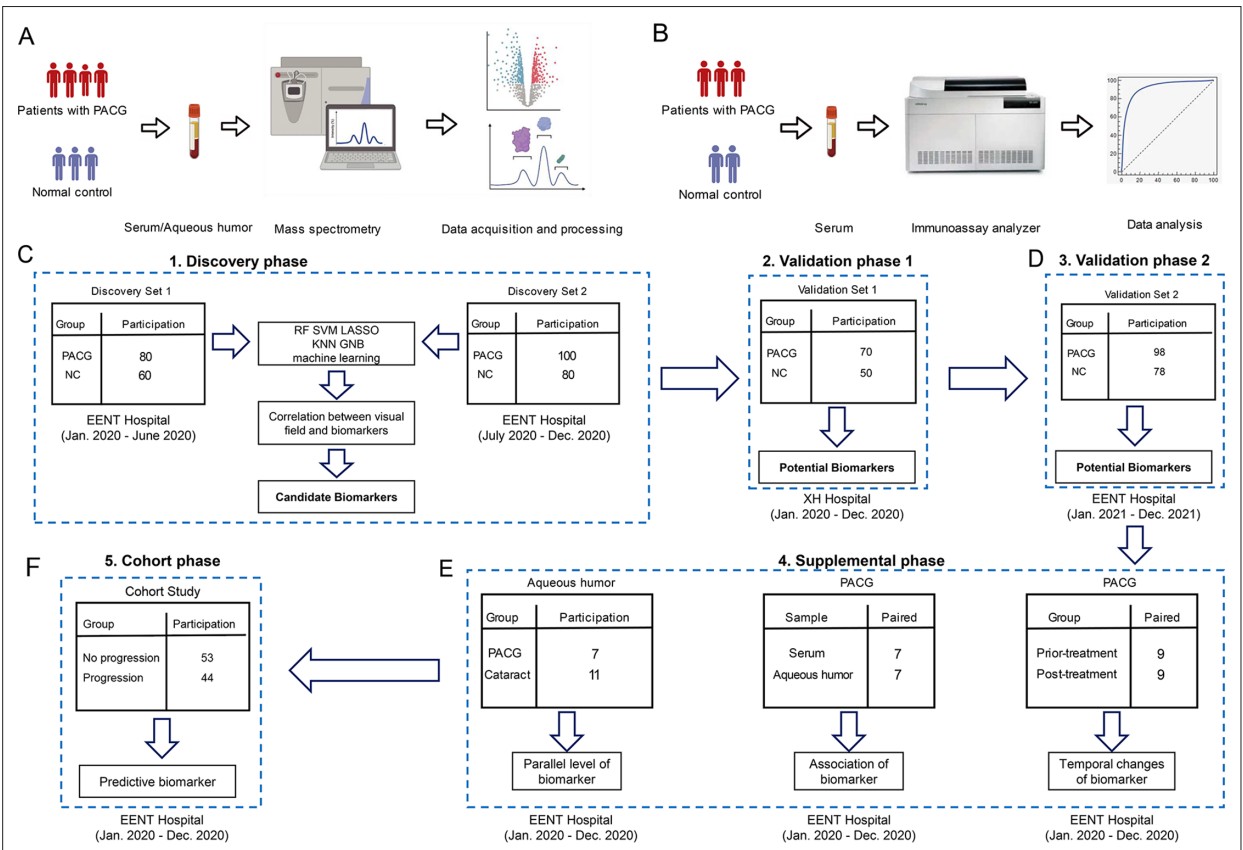

**Figure 1.** Study design and workflow. A five-phase study (discovery phase [discovery set 1, the discovery set 2], validation phase 1, validation phase 2, supplemental phase, and cohort phase) design. (**A**) The workflow of the discovery phase, validation phase 1, supplemental phase, and cohort phase were analyzed by liquid chromatography-mass spectrometry (LC-MS) for untargeted metabolomics. (**B**) The workflow of validation phase 2 was analyzed by chemiluminescence immunoassay for targeted detection. (**C**) A total of 440 patients and controls were recruited and assigned to discovery set 1 (n=140), discovery set 2 (n=180), and validation set 1 (n=120). The biomarker signature was identified on the metabolomic data from the discovery phase, comparing primary angle closure glaucoma (PACG) with control patients. These data were used as a discovery set for the algorithm. (**D**) Validation phase 2 (n=176) was included as the second validation cohort. (**E**) Three measurements were performed in the supplemental phase. (**F**) Cohort phase was performed to validate the predictive value of biomarker (n=97).

The online version of this article includes the following figure supplement(s) for figure 1:

**Figure supplement 1.** Locations of the eight districts in China.

**Figure supplement 2.** The workflow of mass spectrometry.

## Study design

A cross-sectional, multicenter, prospective cohort study design: In the cross-sectional study, a total of 616 patients and controls were prospectively enrolled from the Eye Center of Fudan University and Shanghai Xuhui Central Hospital which was divided into four phases (discovery phase [discovery set 1, the discovery set 2], validation phase 1 [external validation], validation phase 2 [internal validation], supplemental phase) from eight districts in China (*Figure 1—figure supplement 1*). The five phases of the study are independent and are shown in *Figure 1*. The discovery set 1 was composed of 80 serum samples from PACG patients and 60 samples from controls which was enrolled from Eye Center of Fudan University between January 2020 and June 2020. The discovery set 2 was composed of 100 serum samples from PACG patients and 80 samples from controls which was enrolled from Eye Center of Fudan University between July 2020 and December 2020. The differential metabolites of discovery set 1 and discovery set 2 were obtained by widely targeted metabolomics. The intersection of the differential metabolites of the two discovery sets was used as a candidate biomarker. For validation phase 1, serum samples from 70 PACG patients and 50 controls were collected which was enrolled from Shanghai Xuhui Central Hospital between January 2020 and December 2020. Candidate biomarkers were validated in validation phase 1 using widely targeted metabolomics, and potential biomarkers were obtained. For validation phase 2, serum samples from 98 PACG patients and 78 controls were collected which was enrolled from Eye Center of Fudan University between January 2021 and December 2021. Potential biomarkers were validated in validation phase 2 using chemiluminescence methods. For the supplemental phase, we used widely targeted metabolomics to investigate whether the same potential biomarkers were present in PACG patients' aqueous humor. All the clinical characteristics (age, sex, BMI, hypercholesterolemia, hypertension, diabetes, smoking, drinking) were matched between PACG and controls in the four phases.

In the prospective cohort study, 98 newly diagnosed PACG patients were included from the Eye Center of Fudan University between January 2020 and December 2020. All participants visited once every 6 months to allow regular assessment of PACG disease progression (the minimum follow-up period was set to 24 months). Serum samples from 98 PACG patients were collected and measured using chemiluminescence methods. Detailed information of patients' follow-up were described as previously (*Li et al., 2023*; *Li et al., 2020*).

## Sample preparation

### Sample collection

The blood was collected prior to the medical or surgical treatment. Briefly, subjects were asked to ensure about 8–10 hr of fasting before sampling. Venous blood samples were collected in heparinized tubes between 7:00 and 9:30 am. The clinical laboratory obtained the sample at about 10:00 am and centrifuged them at 3000 rpm for 10 min. Then serum was collected into a sterilized cryotube and immediately stored at –80°C for metabolomic analysis.

The acquisition of aqueous humor was described previously (*Tang et al., 2021*). Aqueous humor was collected by skilled ophthalmologists during the surgical treatment of PACG and cataract patients. Aqueous humor was obtained at the start of the procedure using a fine-bore needle during a corneal paracentesis. Following that, aqueous humor samples were quickly transferred to sterile cryotubes and kept at –80°C for metabolomic analysis.

## Sample preprocessing

The same internal standard was added to each sample during metabolite extraction (L-2-chlorophenylalanine, 4-fluoro-L-α-phenylglycine, [2H5]-kynurenic acid, [2H5]-phenoxy acetic acid, indole-3-butyric-2,2-d2 acid, LysoPC 19:0, DL-3-indole-lactic acid). Before extraction, the serum was thawed on ice, vortexed for 10 s, and mixed well. 300 µl of pure cold methanol was added to 50 µl of serum, swirled for 3 min, and centrifuged at 12,000 rpm for 10 min at 4°C. After centrifugation, transfer the supernatant into a new centrifuge tube and place it in a –20°C refrigerator for 30 min. The sample was thawed on ice and centrifuged at 12,000 rpm for 3 min. Transfer 180 µl of the supernatant to the injection vial for mass spectrometry analysis. Quality control (QC) samples are generated by pooling all the serum samples to monitor the retention time and signal intensity consistency. Equal volumes (10 µl) of all serum samples were combined to generate QC samples, which were employed to monitor the repeatability of the analysis. During the mass spectrometry analysis, one QC sample

was included in every 10 samples to ensure the consistency of the analytical process. The PACG and control groups' serum samples were randomly arranged and analyzed. The laboratory conducting the metabolomics measurements was blinded to the samples' case/control/QC status.

## Analytical methods (*Gong et al., 2022*)

### Untargeted metabolomics analysis

Ultra performance liquid chromatography (UPLC) (ExionLC AD, AB SCIEX) separation was performed using Waters ACQUITY HSS T3 (2.1×100 mm$^2$, 1.8 µm). The oven temperature was set to 40°C, and the sample injection volume was 5 µl. Metabolites were eluted from the column at a flow rate of 0.35 ml/min. Mobile phases for UPLC consisted of 0.1% acetic acid in water (phase A) and 0.1% acetic acid in acetonitrile (phase B). The following gradient elution program was employed: 0–10 min: linear gradient from 5% to 90% B; 10–11 min: 90% B; 11–11.1 min: linear gradient from 90% to 5% B; 11.1–14 min: 5% B. Metabolic extracts mixture (QC sample) were analyzed by the triple time of flight (TOF) mass spectrometer (TripleTOF 6600, AB SCIEX) in both positive and negative ionization modes. The scan range was 50–1000 m/z. Electrospray ionization (ESI) source conditions were set as follows: ion spray voltage (IS) 5500 V (positive), –4500 V (negative); ion source gas I (GSI), gas II (GASII), curtain gas (CUR) were set at 50, 50, and 25 psi, respectively; collision energy (CE) 30 V.

### Widely targeted detection conditions

UPLC (ExionLC AD, AB SCIEX) separation was performed using Waters ACQUITY UPLC C18 (2.1×100 mm$^2$, 1.8 um). The oven temperature was set to 40°C, and the sample injection volume was 2 µl. Metabolites were eluted from the column at a flow rate of 0.35 ml/min. Mobile phases for UPLC consisted of 0.1% acetic acid in water (phase A) and 0.1% acetic acid in acetonitrile (phase B). The following gradient elution program was employed: 0–11 min: linear gradient from 5% to 90% B; 11–12 min: 90% B; 12–12.1 min: linear gradient from 90% to 5% B; 12.1–14 min: 5% B. Metabolic extracts of each sample were analyzed by the triple quadrupole-linear ion trap mass spectrometer (QTRAP 6500, AB SCIEX) in both positive and negative ionization modes. ESI source conditions were set as follows: ion spray voltage (IS) 5500 V (positive), –4500 V (negative); ion source gas I (GSI), gas II (GASII), curtain gas (CUR) were set at 50, 50, and 25 psi, respectively. Each ion pair is scanned for detection based on optimized voltage and CE.

### Metabolite profiling

Mixed samples (QC sample) were made and tested by AB Triple TOF 6600 mass spectrometer, the metabolites identified base on public database including Metware public database (Metlin, HMDB, KEGG), and MetDNA. The detected metabolites of QC sample (metabolites with a total score >0.5) add Metware in-house database to be a new whole database. MRM was used for each samples to determine the final ion pair and other information. Based on the new database, QTRAP 6500 was used to quantify all samples accurately. The workflow of mass spectrometry is detailed in *Figure 1—figure supplement 2*.

### Data processing

All LC-MS data were processed using Analyst 1.6.3 for imputing raw, peak picking, alignment, normalization, and to produce peak intensities for retention time and m/z data pages. The features were selected based on their CV with QC samples. Features with CVs of more than 15% were eliminated.

### Chemiluminescence immunoassay

Serum levels of androstenedione were measured using a commercially available kit (Snibe Diagnostics, Shenzhen, China) and were determined using the chemiluminescent immunoassay method by Roche Cobase e 601 (Germany).

### Sample size and missing value

In the cross-sectional study, to calculate the minimum total sample size, we used an open-source calculator based on the methods described by *Obuchowski and Zhou, 2002*, and *Li and Fine, 2004*. The input parameters were specificity = 0.9 (allowable error = 0.05), sensitivity = 0.9 (allowable error

= 0.05), α=0.05 (two-tailed). Based on this calculation, the minimum sample size required for the new biomarker was 98 per phase.

An unreliable conclusion would result from missing data, which would introduce bias. In this study, a total of 48 participants (lack of VF value = 18, lack of medication history record = 12, lack of IOP value = 10, loss to follow up=8) were excluded due to with missing data. Thus no patients with missing data were included in this study.

## Statistical analysis

Normality was assessed using the Shapiro-Wilk W-test. Independent Student's t-test, Kruskal-Wallis test, one-way ANOVA, Wilcox test, and chi-square tests were used when appropriate. Results are presented as frequency and percentage for categorical variables, mean ± SD for normally distributed continuous variables, and median (interquartile range) for not normally distributed continuous variables. The Spearman correlation test was used to determine the significance of the correlations between the variables.

Diagnostic efficiency was evaluated using receiver operating characteristic (ROC) curves. Five machine learning (random forest, support vector machine, lasso, K-nearest neighbor [KNN], and Gaussian Naive Bayes [NB]) approaches were used to identify an optimal algorithm. The Youden index maximizing sensitivity plus specificity is applied to determine the best cutoff value, sensitivity, specificity, accuracy, positive predictive value, and negative predictive value were also calculated. Hosmer-Lemeshow tests were used to assess the goodness of fit. The calibration of the biomarker was assessed by computing the calibration plot. Hanley-McNeil method was used to compare these areas under the receiver operating characteristic curve (AUCs).

Unsupervised principal component analysis (PCA) was performed by statistics function prcomp withe R. The data was unit variance scaled before unsupervised PCA. Supervised orthogonal projections to latent structures-discriminate analysis (OPLS-DA) were applied to obtain a higher level of group separation and a better understanding of variables responsible for classification. Heatmaps of samples and metabolites were carried out by R package (ComplexHeatmap; heatmap). In order to evaluate the binding mode of metabolites and proteins, Autodock Vina v.1.2.2 was used to analyze molecular docking. Fold change = the characteristic peak area of metabolites in the PACG group/the control group.

Identified metabolites were annotated using the KEGG Compound database (http://www.kegg.jp/kegg/compound/). Then annotated metabolites were mapped to the KEGG Pathway database (http://www.kegg.jp/kegg/pathway.html). Significantly enriched pathways were identified with a hypergeometric test's p-value for metabolites.

Cox proportional hazards analysis was also carried out to investigate the relationship between baseline androstenedione levels and VF progression loss. The HRs and baseline characteristics that would assist in categorizing participants into the non-progressing PACG group over the follow-up period were determined using Cox proportional hazards models. Kaplan-Meier plots were used to study the survival results, and the log-rank test was applied to see whether there were any differences between the produced plots.

All statistical analyses were performed using R programming language and SPSS 13.0 (SPSS Inc, Chicago, IL, USA), a list of statistical approaches and packages detailed in *Supplementary file 1*. p-Values <5% were considered statistically significant.

## Results
### Metabolomic analyses in samples from PACG patients and normal controls

The design of this study is depicted in *Figure 1*, while the clinical features of all participants in the four phases of the study are presented in *Table 1* (Phases 1–3), *Supplementary file 2*, and *Supplementary file 3* (Phase 4). All clinical characteristics, including age, sex, BMI, hypercholesterolemia, hypertension, diabetes, smoking, and drinking, were carefully matched between the PACG and normal control groups across all four phases.

Following rigorous QC, data filtering, and normalization procedures, a total of 1464 metabolites were identified across the various samples (N=440). During the discovery phase, the OPLS-DA analysis

**Table 1.** The clinical and demographic characteristics of all subjects in the discovery and validation phases.

| | Normal (n=268) | PACG (n=348) | t/χ² | p |
|---|---|---|---|---|
| **Discovery phase** | | | | |
| **Discovery set 1** | | | | |
| Number (n) | 60 | 80 | | |
| Age (years) | 58.75±8.63 | 61.00±8.67 | 1.52 | 0.13 |
| Sex (male, %) | 24 (40.0) | 31 (38.8) | 0.02 | 0.88 |
| BMI (kg/m²) | 24.04±5.27 | 23.36±2.44 | 0.99 | 0.33 |
| Hypercholesterolemia (yes, %) | 6 (10) | 9 (11.3) | 0.06 | 0.81 |
| Hypertension (yes, %) | 16 (26.7) | 18 (22.8) | 0.32 | 0.60 |
| Diabetes (yes, %) | 1 (1.7) | 8 (10.1) | 3.96 | 0.10 |
| Smoking (yes, %) | 5 (8.3) | 11 (13.9) | 0.99 | 0.42 |
| Drinking (yes, %) | 7 (11.7) | 8 (10.1) | 0.1 | 0.76 |
| Duration (months) | | 10.45±12.43 | | |
| IOP (mmHg) | 12.22±4.50 | 27.78±11.20 | 10.17 | <0.001 |
| VCDR | 0.25±0.18 | 0.64±0.23 | 10.87 | <0.001 |
| AL (mm) | | 22.48±1.14 | | |
| ACD (mm) | | 1.86±0.55 | | |
| CCT (μm) | | 534.04±41.26 | | |
| MS (dB) | | 12.46±8.75 | | |
| MD (dB) | | 14.67±8.94 | | |
| **Discovery set 2** | | | | |
| Number (n) | 80 | 100 | | |
| Age (years) | 62.54±6.74 | 63.14±9.04 | 0.49 | 0.62 |
| Sex (male, %) | 35 (43.8) | 33 (33.0) | 2.19 | 0.14 |
| BMI (kg/m²) | 23.68±2.81 | 23.32±3.15 | 1.36 | 0.18 |
| Hypercholesterolemia (yes, %) | 12 (15) | 13 (13) | 0.15 | 0.70 |
| Hypertension (yes, %) | 26 (32.5) | 33 (33.0) | 0.01 | 0.94 |
| Diabetes (yes, %) | 7 (8.8) | 9 (9.0) | 0.003 | 0.95 |
| Smoking (yes, %) | 11 (13.8) | 15 (15.0) | 0.06 | 0.81 |
| Drinking (yes, %) | 19 (23.8) | 22 (22.0) | 0.08 | 0.78 |
| Duration (months) | | 8.46±9.69 | | |
| IOP (mmHg) | 11.45±5.21 | 25.80±12.55 | 8.329 | <0.001 |
| VCDR | 0.27±0.14 | 0.61±0.22 | 10.48 | <0.001 |
| AL (mm) | | 22.43±0.75 | | |
| ACD (mm) | | 2.05±0.77 | | |
| CCT (μm) | | 547.43±43.04 | | |
| MS (dB) | | 12.34±8.55 | | |
| MD (dB) | | 14.50±8.95 | | |
| **Validation phase 1** | | | | |
| Number (n) | 50 | 70 | | |

*Table 1 continued on next page*

*Table 1 continued*

| **Validation phase 1** | | | | |
|---|---|---|---|---|
| Age (years) | 57.47±8.17 | 60.34±10.11 | 1.66 | 0.10 |
| Sex (male, %) | 18 (36.0) | 24 (34.3) | 0.04 | 0.85 |
| BMI (kg/m²) | 23.08±1.99 | 24.45±3.72 | 1.31 | 0.19 |
| Hypercholesterolemia (yes, %) | 4 (8) | 8 (11.4) | 0.38 | 0.54 |
| Hypertension (yes, %) | 17 (34.0) | 21 (30.0) | 0.22 | 0.64 |
| Diabetes (yes, %) | 5 (10.0) | 7 (10.0) | 0.0 | 1.0 |
| Smoking (yes, %) | 6 (12.0) | 10 (14.3) | 0.13 | 0.72 |
| Drinking (yes, %) | 10 (20.0) | 15 (21.4) | 0.04 | 0.85 |
| Duration (months) | | 11.09±13.82 | | |
| IOP (mmHg) | 12.90±4.11 | 28.46±9.80 | 11.55 | <0.001 |
| VCDR | 0.24±0.17 | 0.60±0.21 | 10.87 | <0.001 |
| AL (mm) | | 22.44±0.83 | | |
| ACD (mm) | | 1.86±0.37 | | |
| CCT (µm) | | 539.09±82.70 | | |
| MS (dB) | | 13.78±8.58 | | |
| MD (dB) | | 13.96±9.29 | | |
| **Validation phase 2** | | | | |
| Number (n) | 78 | 98 | | |
| Age (years) | 56.55±11.53 | 60.26±15.41 | 1.77 | 0.08 |
| Sex (male, %) | 29 (37.2) | 48 (49.0) | 2.46 | 0.12 |
| BMI (kg/m²) | 24.54±5.42 | 25.90±7.51 | 1.26 | 0.21 |
| Hypercholesterolemia (yes, %) | 11 (14.1) | 13 (13.3) | 0.03 | 0.87 |
| Hypertension (yes, %) | 19 (24.4) | 31 (31.6) | 0.26 | 0.61 |
| Diabetes (yes, %) | 6 (7.7) | 17 (17.3) | 3.6 | 0.06 |
| Smoking (yes, %) | 9 (11.5) | 22 (22.4) | 3.6 | 0.06 |
| Drinking (yes, %) | 12 (15.4) | 19 (19.4) | 0.48 | 0.49 |
| Duration (months) | | 11.40±13.8 | | |
| IOP (mmHg) | 13.40±5.43 | 29.20±11.20 | 10.07 | <0.001 |
| VCDR | 0.30±0.19 | 0.68±0.23 | 10.41 | <0.001 |
| AL (mm) | | 23.20±1.72 | | |
| ACD (mm) | | 2.15±0.67 | | |
| CCT (µm) | | 546.71±50.35 | | |
| MS (dB) | | 11.07±8.54 | | |
| MD (dB) | | 16.49±8.64 | | |

BMI = body mass index. IOP = intraocular pressure, VCDR = vertical cup-to-disc ratio, AL = axial length, CCT = central corneal thickness, ACD = anterior chamber depth, MD: visual field mean deviation, MS: visual field mean sensitivity, PACG = primary angle closure glaucoma.

revealed notable distinctions between participants with PACG and those without the condition (*Figure 2A*). Volcano plots (*Figure 2B*) were generated using metabolites exhibiting a false discovery rate (FDR) of <0.1 and fold changes >1.15 or <0.85. In discovery sets 1 and 2, 268 metabolites (21.5%) and 117 metabolites (9.6%), respectively, were found to be significantly altered between PACG and normal subjects.

The Venn diagram depicted in *Figure 2C* illustrates the presence of 32 metabolites that were found to be common in both discovery set 1 (*Supplementary file 4*) and discovery set 2 (*Supplementary file 5*). The parameters used for detecting these 32 differential metabolites are detailed in *Supplementary file 6*. Subsequently, these differential metabolites were utilized for conducting clustering analysis, which is visually represented in the form of a heatmap for both discovery set 1 (*Figure 2D*) and discovery set 2 (*Figure 2E*). These 32 differential metabolites were mainly related to alcohol and amines, amino acid and its metabolomics, benzene and substituted derivatives, fatty acid, heterocyclic compounds, hormones, and hormone-related compounds, nucleotide and its metabolomics, organic acid and its derivatives class (*Figure 2D, E*).

## The blood differential metabolite discriminates PACG from normal

Can these metabolites in blood differentials be considered as potential biomarkers for PACG? In order to investigate this, we computed the AUC for each of the 32 differential metabolites to evaluate their discriminatory capacity in distinguishing PACG from healthy individuals using five machine learning techniques (random forest, support vector machine, lasso, KNN, and Gaussian NB). Among these approaches, KNN demonstrated the highest performance in identifying PACG from normal controls. Subsequently, a two-column heatmap was generated to display the resulting AUC values for discovery set 1 and discovery set 2, respectively, as depicted in *Figure 3A*. The ROC analysis of 32 metabolites revealed AUC values ranging from 0.74 to 1.0 in discovery set 1 and 0.72–1.0 in discovery set 2 for distinguishing PACG from normal subjects.

The present study reveals that the eigenmetabolite levels were markedly elevated ($p < 0.001$) in the normal group compared to the PACG group, as evidenced by the results obtained from discovery set 1 (*Figure 3B*) and discovery set 2 (*Figure 3C*). These findings provide compelling evidence that the heightened risk of PACG is linked to a robust blood metabolite signature.

## Biomarker discovery to discriminate PACG from normal

*Figure 3A* illustrates that 32 metabolites were verified during the discovery phase and were subsequently identified as potential biomarkers. *Figure 4A* depicts a correlation analysis between ocular clinical characteristics and the 32 potential biomarkers discovered during the PACG discovery phase. Notably, a significant positive correlation was observed between androstenedione and the mean deviation of the VF (MD) (r=0.45, $p < 0.001$). This correlation was also observed in discovery set 1 (*Figure 4—figure supplement 1*, r=0.37, $p < 0.001$) and discovery set 2 (*Figure 4—figure supplement 2*, r=0.50, $p < 0.001$), respectively. A statistically insignificant ($p > 0.05$) correlation was observed between ocular clinical characteristics and the remaining 31 potential biomarkers, as depicted in *Figure 4A*, *Figure 4—figure supplement 1*, and *Figure 4—figure supplement 2*. Notably, the level of androstenedione was found to be significantly higher in PACG patients than in normal subjects in both discovery set 1 (*Figure 4B*, p=0.0081, normal: 33,987±11,113, PACG: 42,852±20,767) and discovery set 2 (*Figure 4C*, p=0.0078, normal: 31,559±10,975, PACG:,37,934±18,529). Additionally, high levels of androstenedione were identified as an independent risk factor for PACG, as shown in *Supplementary file 7*. *Figure 3—figure supplement 1* (discovery set 1) and *Figure 3—figure supplement 2* (discovery set 2) illustrate the levels of the remaining 31 metabolites between PACG patients and normal subjects. Based on the preceding analysis, androstenedione was chosen using KNN due to its ability to differentiate between PACG and normal subjects. In discovery set 1, the AUC for PACG versus control was 1.0 (95% CI, 1.0–1.0), as shown in *Figure 4D* and *Supplementary file 8*. Similarly, in discovery set 2, PACG was identified with an AUC of 0.85 (95% CI, 0.80–0.90) when compared to control individuals, as depicted in *Figure 4E* and *Supplementary file 8*.

## Biomarker validation in two independent validation phases

During the validation phases, androstenedione was evaluated as a biomarker signature. The samples from validation phase 1 were subjected to LC-MS analysis for widely targeted metabolomics, which

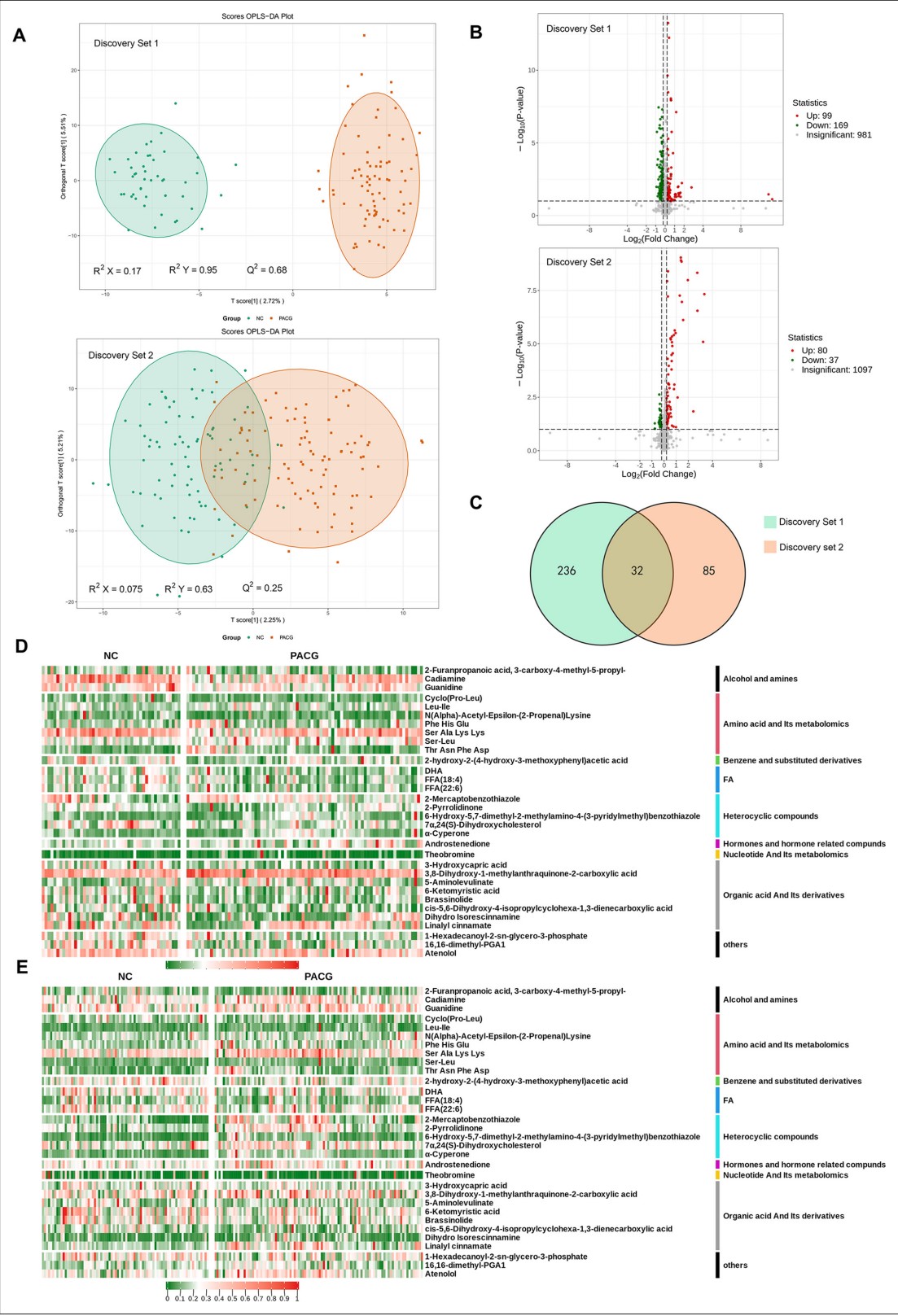

**Figure 2.** Metabolic profiles discriminate participants with primary angle closure glaucoma (PACG) from normal controls (NC). (**A**) Orthogonal projection to latent structure-discriminant analysis (OPLS-DA) score plot of the comparison between the PACG and NC groups in the discovery phase (discovery set 1 and discovery set 2). Samples in the encircled areas are within the 95% confidence interval. (**B**) Volcano plot of differential metabolites. Metabolites with a fold change of <0.85 and a false discovery rate (FDR) of <0.1 were considered significantly down-regulated. Metabolites with a fold change

*Figure 2 continued on next page*

*Figure 2 continued*

of >1.15 and an FDR of <0.1 were considered significantly up-regulated. Changes in other metabolites were not significant (insignificant). (**C**) Venn diagram displaying the 32 differential metabolites that were altered as biomarker candidates from the two comparisons in the discovery phase. (**D**) Heatmap of differential metabolites in the discovery set 1 (data were normalized to min-max). (**E**) Heatmap of differential metabolites in the discovery set 2 (data were normalized to min-max). FA: fatty acid.

revealed significant differences between PACG and normal participants, as demonstrated by the OPLS-DA (*Figure 5—figure supplement 1*). In validation phase 2, a chemiluminescence immuno-assay method was developed to enable precise quantification of serum androstenedione levels in a convenient and rapid manner.

During validation phases 1 and 2, the level of androstenedione was found to be significantly higher in individuals with PACG compared to normal subjects (*Figure 5A*, p=0.0042, normal: 60,737±28,078, PACG: 82,394±33,994; *Figure 5B*, p=0.0034, normal: 1.552±0.489, PACG: 1.825±0.6876). The performance of androstenedione was evaluated using the AUC in ROC analysis. The AUC for PACG versus control was 0.87 (95% CI, 0.80–0.95) in validation phase 1, as depicted in *Figure 5C* and *Supplementary file 8*. In validation phase 2, a consistent performance of androstenedione (AUC, 0.86, 95% CI, 0.81–0.91) was observed (*Figure 5D*, *Supplementary file 8*).

## Biomarker validation in male and female subgroups

Validation of androstenedione as a biomarker for PACG necessitates consideration of sex, as males exhibit 5–10 times higher levels of this hormone than females. Accordingly, the subjects were stratified into male and female subgroups.

*Figure 5—figure supplement 2* demonstrates that in the discovery phase, the AUC for PACG compared to control was 1.0 (95% CI, 1.0–1.0) in the male subgroup (*Figure 5—figure supplement 2A*), and 0.91 (95% CI, 0.85–0.98) in the female subgroup (*Figure 5—figure supplement 2B*). In validation phase 1, the AUC for PACG versus control was 0.83 (95% CI, 0.73–0.92) in the male subgroup (*Figure 5—figure supplement 2C*), and 1.0 (95% CI, 1.0–1.0) in the female subgroup (*Figure 5—figure supplement 2D*). In validation phase 2, the AUC for PACG versus control was 1.0 (95% CI, 1.0–1.0) in the male subgroup (*Figure 5—figure supplement 2E*), and 0.88 (95% CI, 0.79–0.98) in the female subgroup (*Figure 5—figure supplement 2F*).

## Androstenedione associates with severity of PACG

The present study aimed to examine the potential association between serum androstenedione levels and the clinical severity of PACG. PACG severity was categorized into mild (MD≤6), moderate (6–12), and severe (MD >12) based on the MD value. The results from both discovery set 1 (*Figure 6A*, mild: 32,600±17,011, moderate: 33,215±17,855, severe: 46,060±21,789) and discovery set 2 (*Figure 6B*, mild: 27,866±19,873, moderate: 27,057±13,166, severe: 43,972±19,234) indicated that the mean serum androstenedione levels were significantly higher in the severe PACG group compared to the moderate and mild PACG groups (p<0.001). These findings were further validated in both validation phase 1 (*Figure 6C*, mild: 75,726±45,719, moderate: 65,798±30,610, severe: 94,348±30,858) and validation phase 2 (*Figure 6D*, mild: 1.121±0.3143 ng/ml, moderate: 1.461±0.4391 ng/ml, severe: 2.147±0.6476 ng/ml).

A correlation analysis was conducted to examine the relationship between MD and androstenedione. Notably, a statistically significant positive correlation was found between MD and androstenedione in both discovery set 1 (*Figure 4—figure supplement 1*, r=0.37, p<0.001) and discovery set 2 (*Figure 4—figure supplement 2*, r=0.50, p<0.001). This finding was further confirmed in validation phase 2 (*Figure 6E*, r=0.61, p<0.001). Subsequently, the diagnostic potential of androstenedione in distinguishing the severity of PACG was investigated. An AUC was calculated for each severity of PACG to evaluate the discriminatory accuracy of androstenedione in distinguishing between mild, moderate, and severe cases. The results of ROC analysis indicated AUC values ranging from 0.75 to 0.95 in discovery set 1 (*Figure 6—figure supplement 1A*, *Supplementary file 8*) and 0.94–0.99 in discovery set 2 (*Figure 6—figure supplement 1B*, *Supplementary file 8*) when using KNN machine learning algorithms. The validity of androstenedione in distinguishing the severity of PACG was demonstrated through consistent performance in both validation phase 1 (*Figure 6—figure supplement 1C*, *Supplementary file 8*, AUC of 0.64–0.97) and validation phase 2 (*Figure 6—figure supplement*

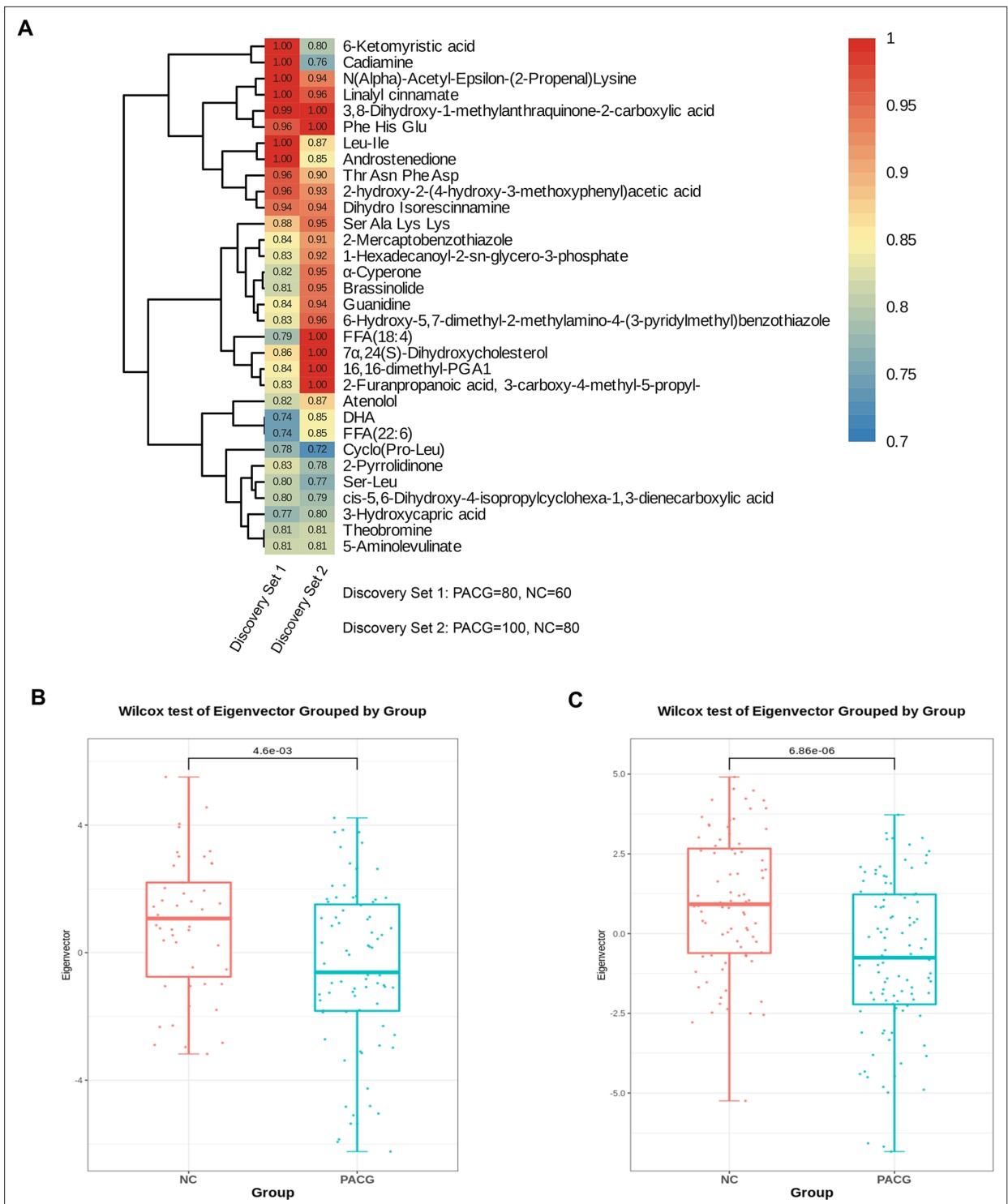

**Figure 3.** Identification of a unique primary angle closure glaucoma (PACG)-associated blood metabolite fingerprint and its behavior in the discovery phase. (**A**) Heatmap of the area under the receiver operating characteristic curve assessing the discriminating accuracy of each of the 32 metabolites in differentiating PACG from normal control in the discovery set 1 and discovery set 2. (**B**) The eigenmetabolite of the 32-metabolite cluster between PACG and control patients in the discovery set 1. (**C**) The eigenmetabolite of the 32-metabolite cluster between PACG and control patients in the discovery set 2. Wilcox test was used.

The online version of this article includes the following figure supplement(s) for figure 3:

*Figure 3 continued on next page*

*Figure 3 continued*

**Figure supplement 1.** The level of the 31-metabolite cluster between primary angle closure glaucoma (PACG) and control patients in discovery set 1 (unit for y-axis is peak areas).

**Figure supplement 2.** The level of the 31-metabolite cluster between primary angle closure glaucoma (PACG) and control patients in discovery set 2 (unit for y-axis is peak areas).

*1D*, *Supplementary file 8*, AUC of 0.98–1.0). Additionally, when mild and moderate cases were combined, the AUC for mild and moderate versus severe was 0.94 (95% CI 0.89–0.99) in discovery set 1 (*Figure 6F*, *Supplementary file 8*), 0.93 (95% CI 0.88–0.98) in discovery set 2 (*Figure 6G*, *Supplementary file 8*), 0.92 (95% CI 0.85–0.99) in validation phase 1 (*Figure 6H*, *Supplementary file 8*), and 0.98 (95% CI 0.96–1.0) in validation phase 2 (*Figure 6I*, *Supplementary file 8*).

## Clinical value of androstenedione in patients with PACG

The lack of specificity of serum biomarkers remains a significant obstacle to the clinical application of such markers. To investigate temporal changes in androstenedione levels during the initial diagnosis and post-treatment period, we conducted a random analysis of nine pairs of blood samples from patients taken before and 3 months after surgical treatment (*Figure 7A*). Our findings indicate a significant decrease in androstenedione levels in the post-treatment serum of the nine patients compared to the pre-treatment serum (p=0.021, *Figure 7B*).

A case-control study (PACG = 7, cataract = 11) (*Figure 7C*) was conducted to investigate the potential elevation of aqueous humor levels of androstenedione in patients. The findings revealed a statistically significant increase (p=0.011) in the levels of androstenedione in the aqueous humor of patients with PACG compared to those with cataracts (*Figure 7D*). Additionally, a significantly positive correlation between MD and aqueous humor levels of androstenedione was observed (*Figure 7E*, r=0.98, p<0.001). The mean aqueous humor levels of androstenedione were found to be significantly higher (p<0.001) in the severe PACG group compared to the moderate and mild PACG groups (*Figure 7F*). Subsequently, an examination was conducted on seven paired serum-aqueous humor samples obtained from identical PACG patients to ascertain the presence of a consistent pattern within the same individuals. A statistically significant correlation was observed between the levels of androstenedione in serum and aqueous humor (r=0.82, p=0.038) (*Figure 7G*).

## Calibration ability of androstenedione on discovery phase and validation phase

*Figure 8* displays the calibration plots for both the discovery and validation phases, which effectively validate the calibration performance of serum androstenedione for probability. The plots demonstrate a high level of agreement between predicted and observed values in both discovery set 1 (*Figure 8A*) and discovery set 2 (*Figure 8B*). The Hosmer-Lemeshow test yielded a nonsignificant statistic in the discovery set 1 ($\chi^2$=0, p=1) and discovery set 2 ($\chi^2$=6.16, p=0.10), indicating no departure from a perfect fit. Validation phase 1 (*Figure 8C*, $\chi^2$=5.14, p=0.14) and validation phase 2 (*Figure 8D*, $\chi^2$=1.25, p=0.26) resulted in similar performance.

## Androstenedione could predict VF progression in patients with PACG

This study comprised 97 participants diagnosed with PACG, selected based on the screening criteria and followed up for a period of 24 months. In cases of bilateral PACG, one eye was selected at random. Of the total participants, 44 (45.36%) exhibited glaucoma progression, as evidenced by VF loss. The demographic and ocular features of the VF progressing and non-progressing groups at baseline are presented in *Supplementary file 9*.

The patients in the progression group exhibited a statistically significant increase (p<0.001) in the mean serum levels of androstenedione compared to those in the non-progressing group (*Supplementary file 9*). Furthermore, the multivariate Cox analysis revealed that the baseline levels of androstenedione (HR = 2.71, 95% CI = 1.20–6.10, p=0.017) were significantly associated with glaucoma progression, as determined by the VF loss results (*Table 2*). *Figure 9* displays the Kaplan-Meier survival curves. The results of the survival analysis revealed a statistically significant increase in the proportion of patients with elevated androstenedione levels who experienced VF progression in PACG (log-rank

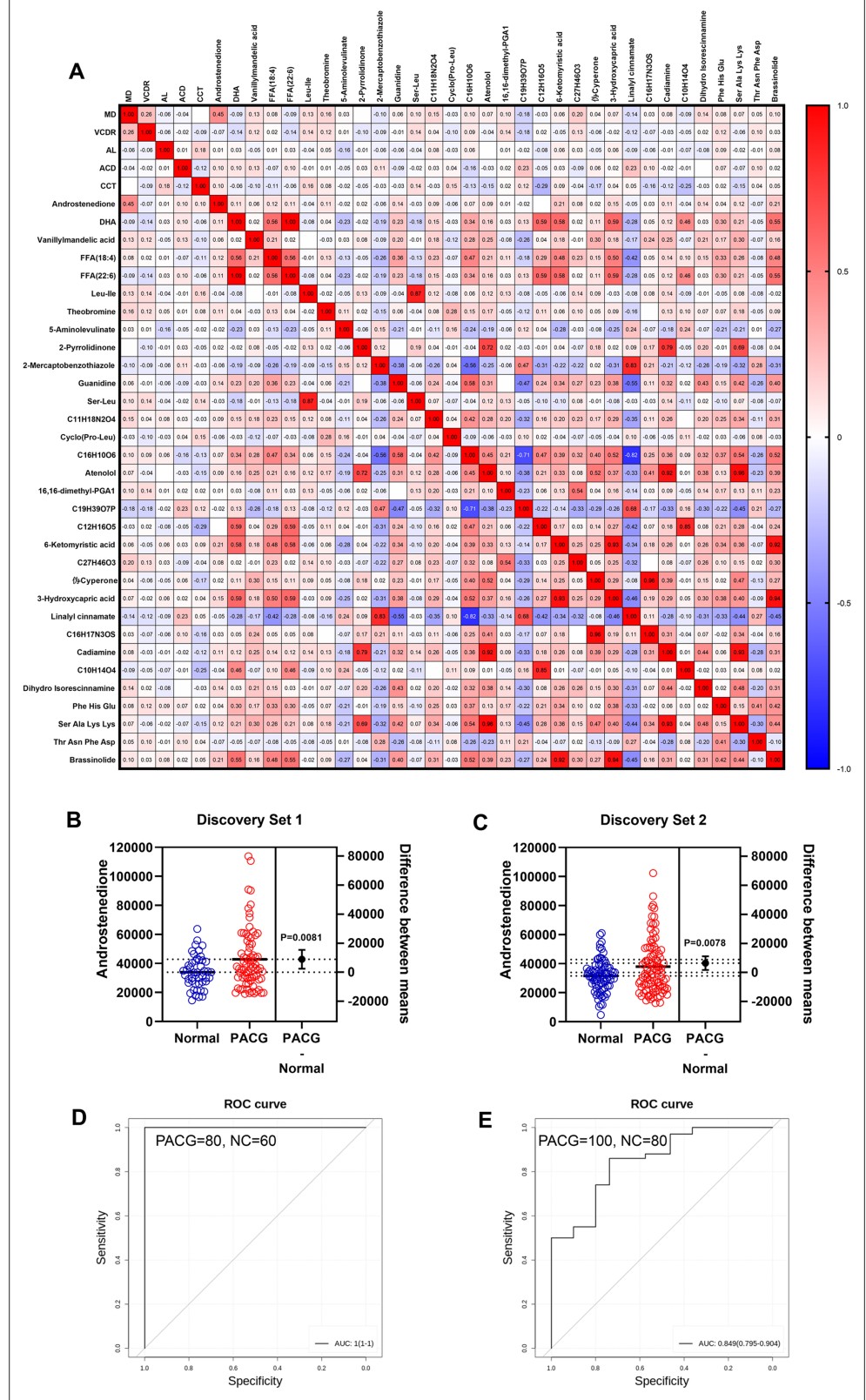

**Figure 4.** Biomarker discovery discriminates primary angle closure glaucoma (PACG) from normal in the discovery phase. (**A**) A heatmap of correlation analysis between ocular clinical characteristics and 32 potential biomarkers in the discovery phase in PACG subjects. (**B**) The serum level of androstenedione between PACG (42,852±20,767) and normal (33,987±11,113) group in the discovery set 1 (unit for y-axis is peak areas). (**C**) The serum level of

*Figure 4 continued on next page*

*Figure 4 continued*

androstenedione between PACG and normal group in the discovery set 2 (unit for y-axis is peak areas). (**D**) Receiver operating characteristic curves of androstenedione to discriminate PACG from normal in the discovery set 1. (**E**) Receiver operating characteristic curves of androstenedione to discriminate PACG from normal in the discovery set 2. Independent Student's t-test was used.

The online version of this article includes the following figure supplement(s) for figure 4:

**Figure supplement 1.** Heatmap of correlation analysis between ocular clinical characteristics and 32 potential biomarkers in discovery set 1 of primary angle closure glaucoma (PACG) subjects.

**Figure supplement 2.** Heatmap of correlation analysis between ocular clinical characteristics and 32 potential biomarkers in discovery set 2 of primary angle closure glaucoma (PACG) subjects.

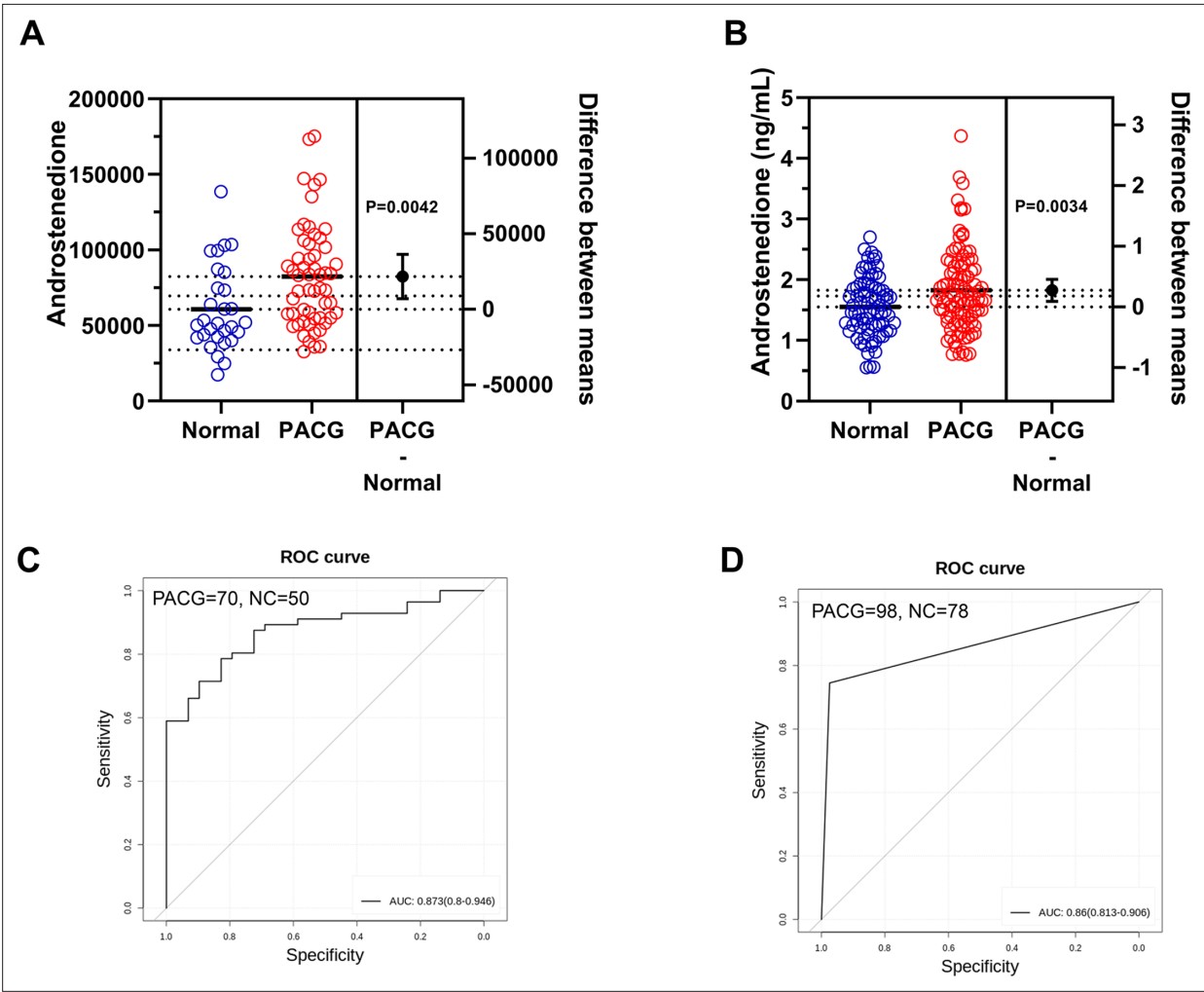

**Figure 5.** Biomarker validation in two independent validation phases to discriminate primary angle closure glaucoma (PACG) from normal. (**A**) The serum level of androstenedione between PACG and normal group in validation phase 1 (unit for y-axis is peak areas). (**B**) The serum level of androstenedione between PACG and normal group in validation phase 2 (unit for y-axis is peak areas). (**C**) Receiver operating characteristic curves of androstenedione to discriminate PACG from normal in validation phase 1. (**D**) Receiver operating characteristic curves of androstenedione to discriminate PACG from normal in validation phase 2. Independent Student's t-test was used.

The online version of this article includes the following figure supplement(s) for figure 5:

**Figure supplement 1.** Orthogonal projection to latent structure-discriminant analysis (OPLS-DA) score plot of the comparison between the primary angle closure glaucoma (PACG) and normal control (NC) groups in validation phase 1.

**Figure supplement 2.** Biomarker validation in male and female subgroups.

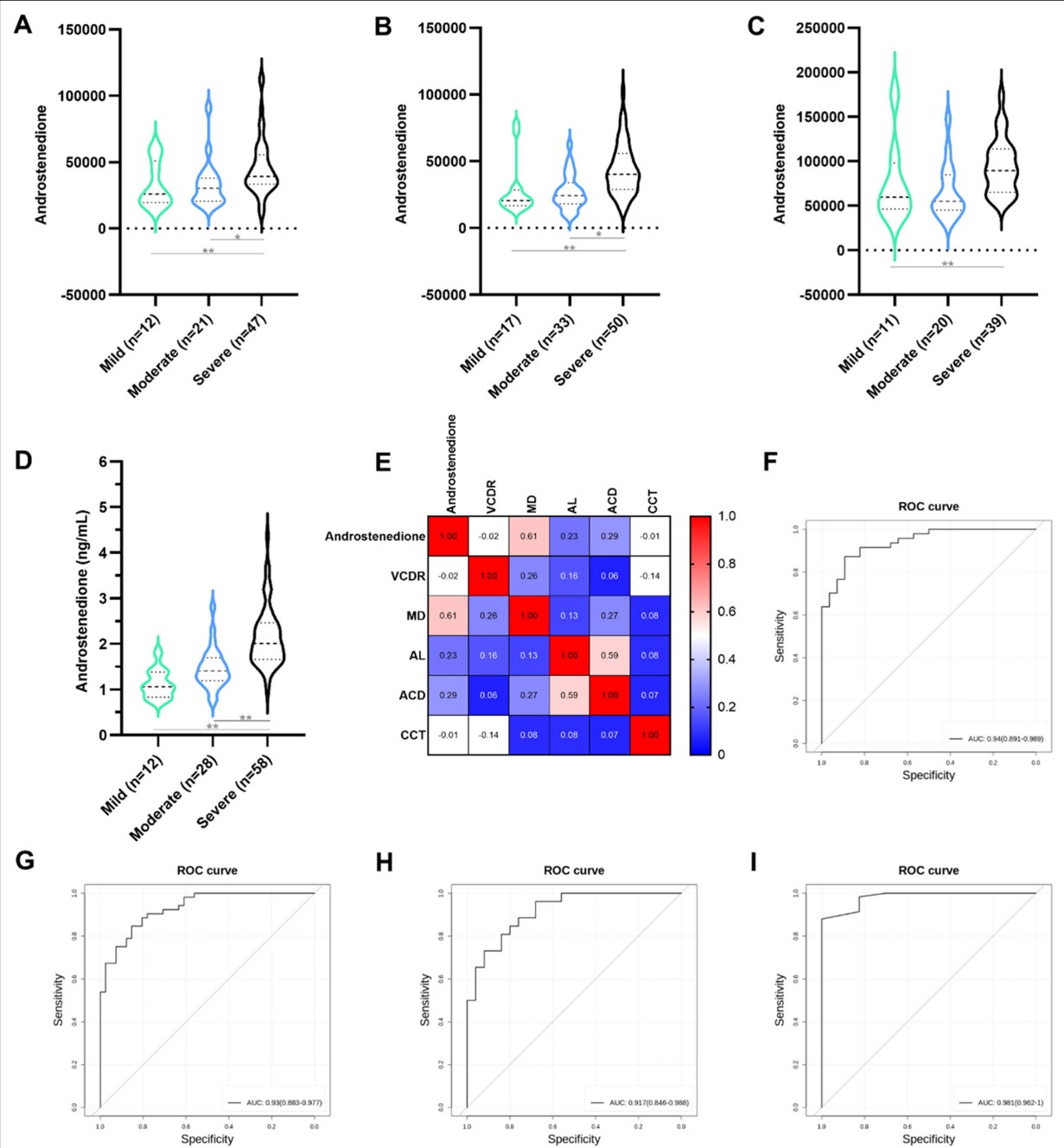

**Figure 6.** Androstenedione is associated with the severity of primary angle closure glaucoma (PACG). (**A**) Comparison of mean serum levels of androstenedione between mild, moderate, and severe PACG in the discovery set 1 (unit for y-axis is peak areas). (**B**) Comparison mean serum levels of androstenedione between mild, moderate, and severe PACG in the discovery set 2 (unit for y-axis is peak areas). (**C**) Comparison of mean serum levels of androstenedione between mild, moderate, and severe PACG in validation phase 1 (unit for y-axis is peak areas). (**D**) Comparison of mean serum levels of androstenedione between mild, moderate, and severe PACG in validation phase 2 (unit for y-axis is peak areas). (**E**) Heatmap of correlation analysis between ocular clinical characteristics and androstenedione in validation phase 2. (**F**) Androstenedione to discriminate mild and moderate PACG from severe PACG in the discovery set 1. (**G**) Androstenedione to discriminate mild and moderate PACG from severe PACG in the discovery set 2. (**H**) Androstenedione to discriminate mild and moderate PACG from severe PACG in validation phase 1. (**I**) Androstenedione to discriminate mild and moderate PACG from severe PACG in validation phase 2. *: p<0.05. **: p<0.001.

The online version of this article includes the following figure supplement(s) for figure 6:

**Figure supplement 1.** Receiver operating characteristic curves to discriminates mild, moderate, and severe primary angle closure glaucoma (PACG).

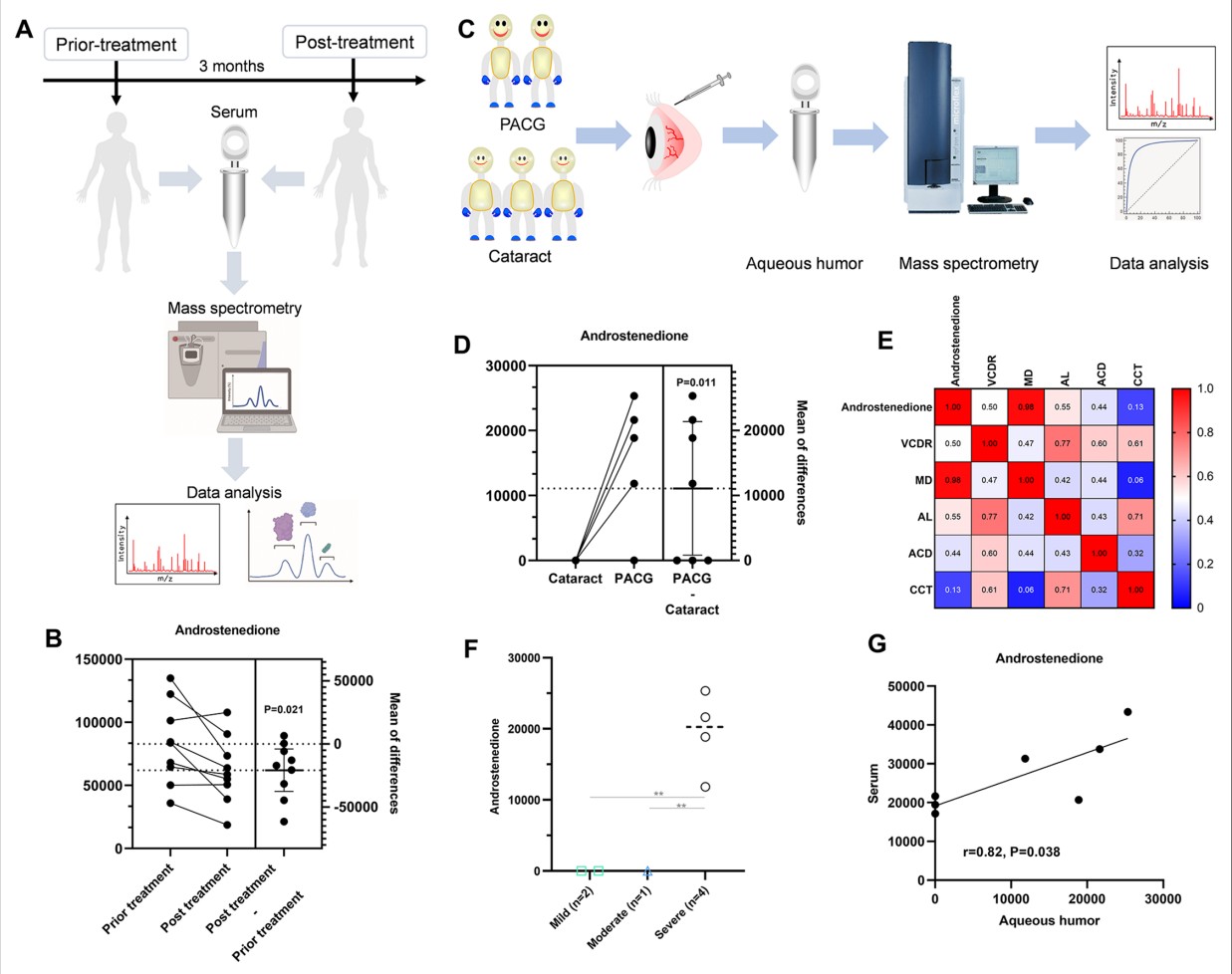

**Figure 7.** Specificity of circulating androstenedione in patients with primary angle closure glaucoma (PACG) in supplemental phase. (**A**) Sampling scheme and workflow to investigate the temporal changes in androstenedione levels. (**B**) Differential level of serum androstenedione between patients with PACG before and 3 months after treatment (unit for y-axis is peak areas). (**C**) Sampling scheme and workflow to determine whether aqueous humor levels of androstenedione were high in patients with PACG. (**D**) The aqueous humor level of androstenedione between PACG and cataract (unit for y-axis is peak areas). (**E**) Heatmap of correlation analysis between ocular clinical characteristics and aqueous humor level of androstenedione. (**F**) Comparison means aqueous humor levels of androstenedione between mild, moderate, and severe PACG (unit for y-axis is peak areas). (**G**) Seven paired serum-aqueous humor samples from the same PACG patients were included (unit for y- and x-axis is peak areas). A significant correlation between serum and aqueous humor levels of androstenedione was observed. Kruskal-Wallis test and one-way ANOVA was used. *: p<0.05; **: p<0.001.

test, p<0.001, *Figure 9A*). Comparable findings were observed in both the female (log-rank test, p=0.0042, *Figure 9B*) and male (log-rank test, p=0.0014, *Figure 9C*) subgroups.

## Discussion

The delayed identification of PACG is a notable contributor to patients' impaired vision and irreversible blindness. Regrettably, existing medical protocols do not advocate for the use of blood-based biomarkers to diagnose and prognosticate PACG. Consequently, it is imperative to develop novel, uncomplicated, and practicable approaches to improve the early detection of PACG and its predictive accuracy. To address the significant challenge at hand, the present study conducted a cross-sectional and prospective cohort investigation utilizing high-throughput, widely targeted metabolomics and targeted chemiluminescence immunoassay on blood samples obtained from a substantial number of patients diagnosed with PACG and healthy individuals. The findings of this study reveal, for the first time, that androstenedione exhibits a reasonable level of precision (AUC, 0.85–1.0) in distinguishing

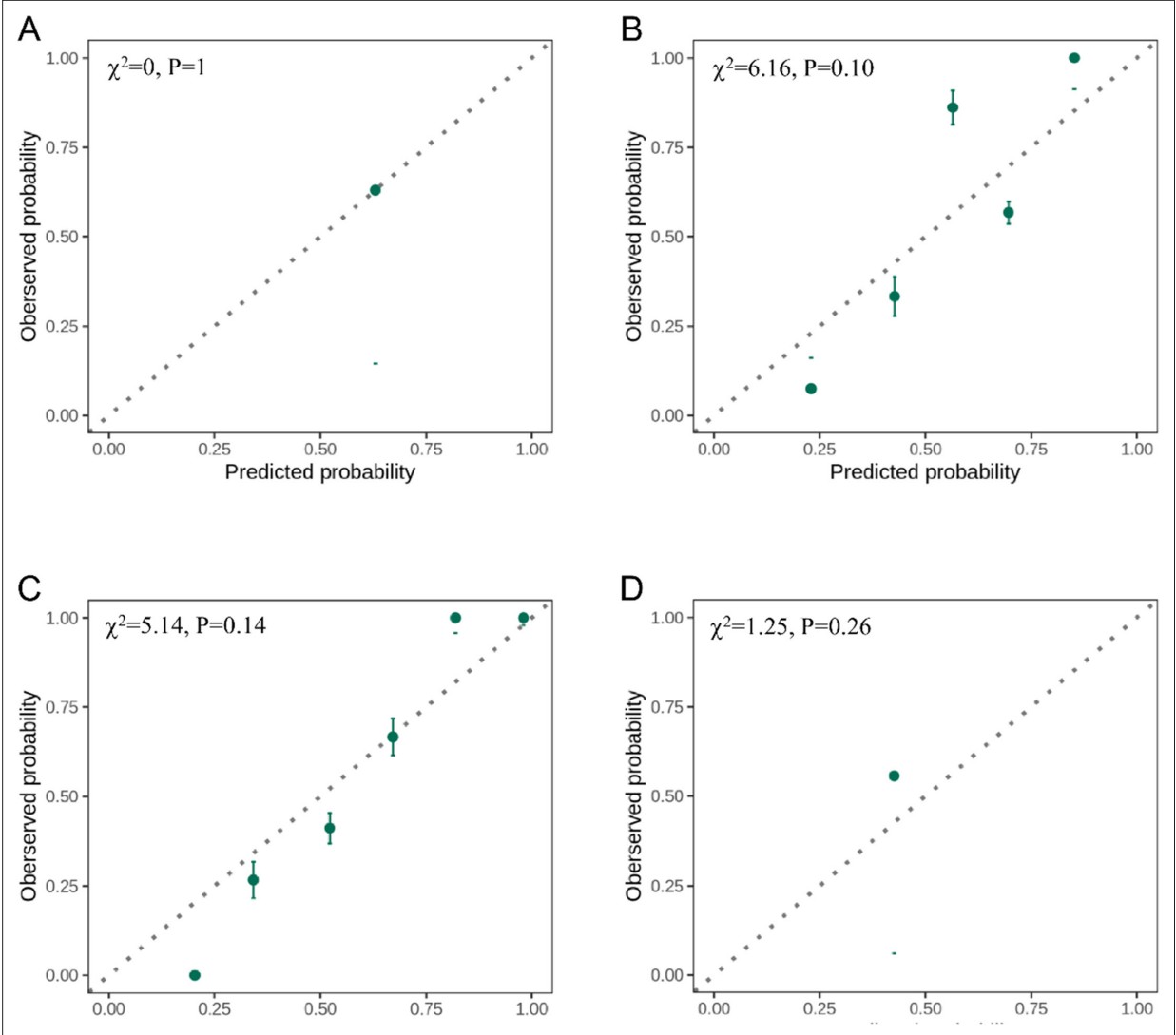

**Figure 8.** Calibration ability of androstenedione. (**A**) Calibration curves in the discovery set 1. (**B**) Calibration curves in the discovery set 2. (**C**) Calibration curves in the validation phase 1. (**D**) Calibration curves in the validation phase 2. The predicted probability of primary angle closure glaucoma (x-axis) is compared to the observed frequency (y-axis). The plot is grouped by deciles and quintiles of the predicted risk (green circles) with 95% confidence intervals (green lines). Perfect predictions should lie on the reference (dashed).

between PACG and control groups in blood samples. Additionally, the baseline levels of androstenedione may serve as a valuable predictor of glaucomatous VF progression.

Some metabolites have been proposed to have a potential role in discriminating PACG from normal, but validation studies in larger cohorts still need to be completed. *Rong et al., 2017*, conducted a case-control (PACG = 38, normal = 48) study using gas chromatography-mass spectrometry and reported that palmitoleic acid, linoleic acid, γ-linolenic acid, and arachidonic acid were identified as essential metabolites associated with PACG, but diagnostic accuracy is unknown. In our study, these metabolites were also detected, but diagnosis accuracy (AUC<0.7) was limited. *Qin et al., 2022*, measured plasma 22 free fatty acids (FFA) and 6 lipid classes using metabolomics analysis, shown that docosahexaenoic acid (DHA) and total saturated fatty acids may be screening indices (AUC, 0.82–0.85) for PACG patients but lack validation set to confirm the results. DHA (AUC, 0.74–0.85), FFA (22:6) (AUC, 0.74–0.85), and FFA (18:4) (AUC, 0.79–1.0) were also shown a diagnose value for PACG patients in the discovery set of our study (*Figure 3A*), but the robustness and diagnose accuracy was weaker than androstenedione (AUC, 0.85–1.0) (*Supplementary file 10*). The contribution of previous

**Table 2.** Cox proportional hazards regression analysis to assess the value of androstenedione associated with progression of PACG.

| | Univariate | | Multivariate* | |
|---|---|---|---|---|
| | p | HR (95% CI) | p | HR (95% CI) |
| Age | **0.008** | **0.97 (0.95–0.99)** | 0.22 | 0.98 (0.95–1.01) |
| Sex | 0.87 | 0.95 (0.53–1.72) | 0.73 | 1.15 (0.53–2.52) |
| IOP | 0.59 | 1.007 (0.98–1.04) | 0.75 | 0.99 (0.96–1.03) |
| VCDR | 0.17 | 0.40 (0.11–1.46) | 0.16 | 0.37 (0.092–1.46) |
| CCT | 0.77 | 1.001 (0.10–1.007) | 0.90 | 1.000 (0.99–1.007) |
| ACD | **0.03** | **1.63 (1.05–2.52)** | 0.91 | 1.041 (0.51–2.13) |
| AL | **0.005** | **1.24 (1.07–1.44)** | 0.35 | 1.12 (0.89–1.41) |
| MD | 0.13 | 1.04 (0.99–1.08) | 0.075 | 1.047 (0.10–1.10) |
| Androstenedione | **<0.001** | **3.73 (1.84–7.57)** | **0.017** | **2.71 (1.20–6.10)** |

*Adjusted for BMI, diabetes (yes = 1, no = 0), hypertension (yes = 1, no = 0), hypercholesterolemia (yes = 1, no = 0), smoking (yes = 1, no = 0), and drinking (yes = 1, no = 0). Bold values indicate positive results.
IOP =intraocular pressure, VCDR =vertical cup-to-disc ratio, AL =axial length, CCT =central corneal thickness, ACD =anterior chamber depth, MD =mean deviation, PACG =primary angle closure glaucoma.

metabolites to PACG diagnosis was small, which prompted us to conduct a trial designed to develop biomarkers.

In this study, androstenedione levels achieved better diagnostic efficiency and robust calibration across the discovery and validation sets. The transformation of normal subjects into PACG is a chronic stepwise process. Given this, screening healthy individuals and those with a family history of the disease is vital. Another important finding of this research was that the level of serum androstenedione retained its diagnostic efficiency for distinguishing the severity of PACG. Furthermore, serum androstenedione levels incrementally increased from mild to moderate to severe PACG, suggesting that serum androstenedione levels may accurately reflect the progression/severity of PACG. Furthermore, androstenedione levels at baseline were a new biomarker for predicting glaucomatous VF progression. Hence, serum androstenedione levels may provide a new biomarker for early detection and monitoring/predicting PACG severity/progression.

In the supplemental phase, we asked whether the serum levels of androstenedione were significantly correlated with aqueous humor levels of androstenedione from the same PACG patients, which is an important criterion for application as a routine clinical biomarker and to explore the biological function in PACG. Fortunately, the levels of aqueous humor androstenedione were positively correlated (r=0.82, p=0.038) with serum levels. Meanwhile, the levels of aqueous humor androstenedione incrementally increased from mild-moderate to severe PACG. Furthermore, the levels of aqueous humor androstenedione remained higher in PACG than in cataracts. Together, these results confirm that androstenedione can serve as a novel biomarker for early detection and monitoring of the progression/severity of PACG.

In clinical practice, oral topical glaucoma medications and laser peripheral iridectomy were the primary therapies used to treat PACG. However, the effects of these therapies are mainly estimated by measurement of IOP and optical coherence tomography. Thus, we have wondered whether, by dynamic detection, serum androstenedione levels could reflect the therapeutic effect of these therapies. In the supplemental phase (fourth phase), in a small cohort of nine PACG patients treated with laser peripheral iridectomy or oral topical glaucoma medications, which showed a better prognosis, seven of nine exhibited decreased levels of serum androstenedione. These results reveal a new biomarker for monitoring the effect of anti-glaucoma therapy; however, they should be verified in a larger cohort.

The main strength of our study is its robustness. We conducted a five-phase study (discovery set [discovery set 1, the discovery set 2], validation phase 1, validation phase 2, supplemental phase, and cohort phase). We used large and well-characterized patients with adequate controls to confirm the

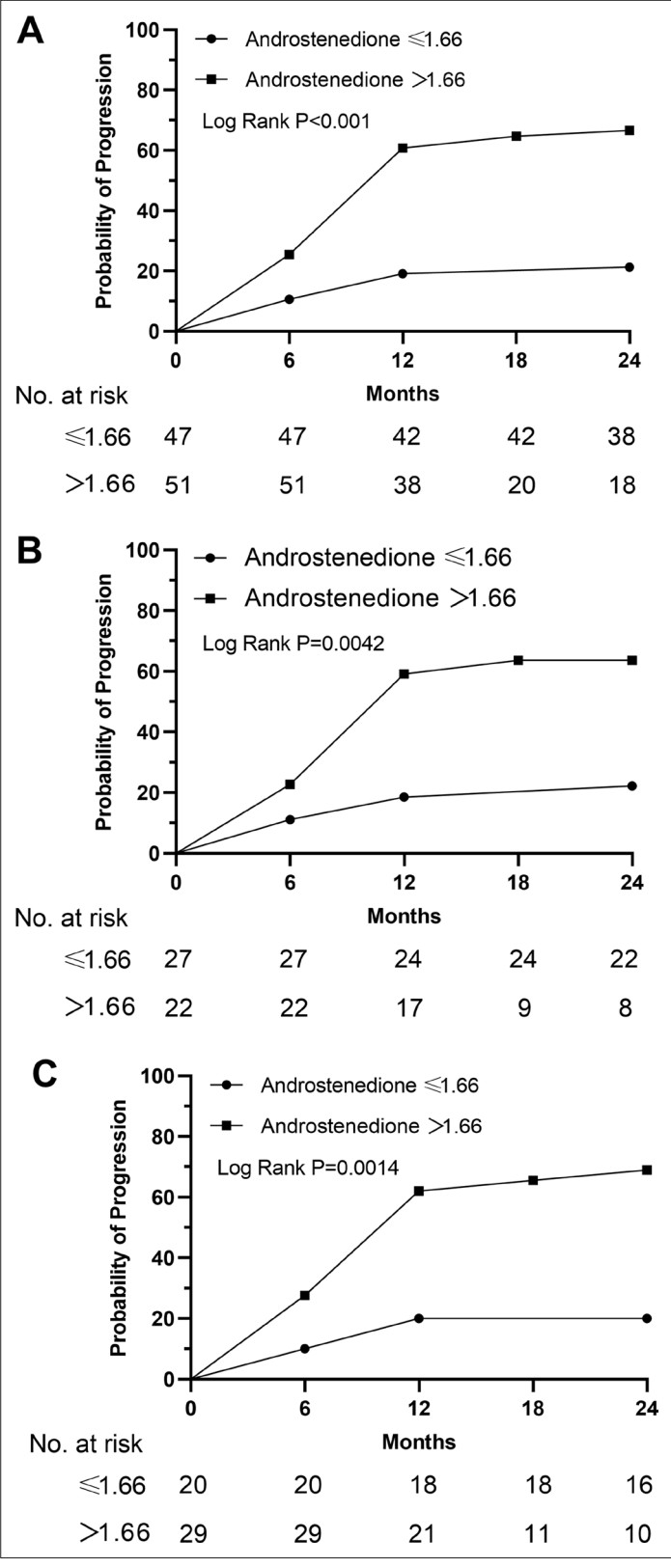

**Figure 9.** Kaplan-Meier curves stratified by the men value in terms of androstenedione. (**A**) Male+female; (**B**) female; (**C**) male. We categorized study participants into two groups based on their mean levels of androstenedione.

results. Widely targeted metabolomics and target chemiluminescence immunoassay methods provide high reliability of the metabolite. Androstenedione achieved better diagnostic accuracy across the discovery and validation sets, with AUC varying between 0.85 and 1.0. Interestingly, baseline androstenedione levels can predict glaucoma progression via VF loss results. Of note, the clinical practice of androstenedione in patients with PACG was analyzed by supplemental phase.

The metabolites identified by this study match PACG pathophysiological concepts. Pathway enrichment analysis demonstrated that these 32 significantly altered metabolites primarily belonged to 16 pathways (*Figure 10—figure supplement 1*). Among the top altered pathways, steroid hormone biosynthesis appears to be the critical node to high-match PACG pathophysiological concepts (*Madjedi et al., 2022*; *Prokai-Tatrai et al., 2013*): it connects with androstenedione. The molecular formula of androstenedione is shown in *Figure 10—figure supplement 2*. Steroid hormone biosynthesis appears to be a key node in the pathophysiological concept of highly matched PACG, but high enrichment is observed in metabolic pathways. Hormones play critical roles in various physiological functions, such as metabolism, immune responses, and inflammation regulation. In a related study on fatigue experienced during androgen deprivation therapy, marked disparities were found in metabolite levels in the steroid hormone biosynthesis pathways,

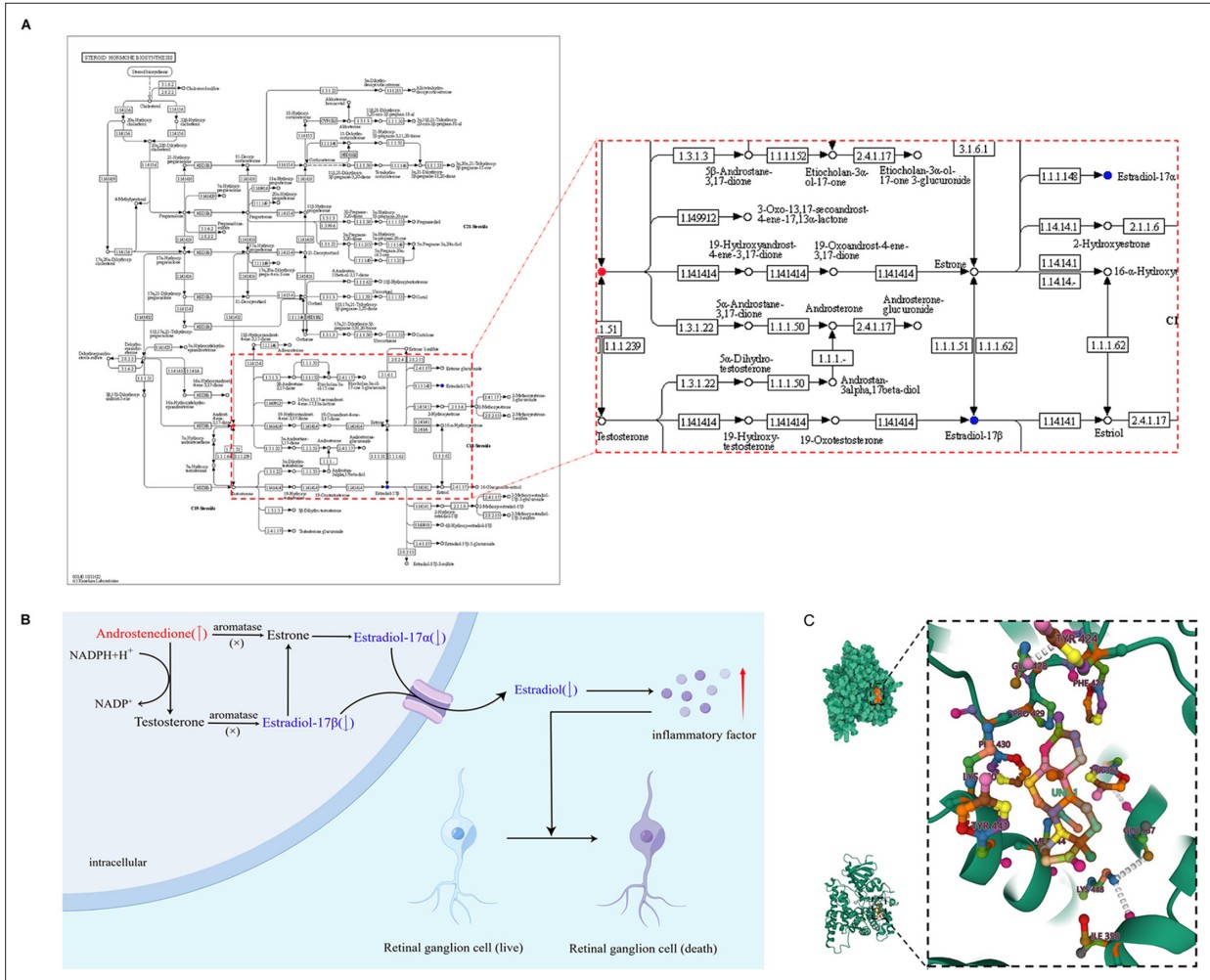

**Figure 10.** Potential mechanisms, androstenedione accumulation in patients with primary angle closure glaucoma (PACG). (**A**) Steroid hormone biosynthesis related to androstenedione. (**B**) Aromatase function deficits may be the potential mechanism leading to androstenedione accumulation in patients with PACG. (**C**) Binding mode of androstenedione to aromatase by molecular docking.

The online version of this article includes the following figure supplement(s) for figure 10:

**Figure supplement 1.** Pathway enrichment plot in discovery set 2.

**Figure supplement 2.** Molecular formula of androstenedione.

underscoring their significance in metabolic shifts. These insights highlight the intricate interconnection between steroid hormone biosynthesis and other metabolic pathways (*Feng et al., 2021*; *Schiffer et al., 2019*). The sex hormones pathway is a vital component of steroid hormone biosynthesis (*Figure 10A*). Several studies have focused on the importance of 17β-estradiol in protecting the retinal ganglion cell layer and preserving visual function in clinical (*Vajaranant et al., 2018*) and basic (*Prokai-Tatrai et al., 2013*) research by anti-inflammatory effect (*Engler-Chiurazzi et al., 2017*). Our previous studies suggested that decreased sex hormone concentrations in glaucoma lead to a hyperinflammatory state, leading to faster rates of VF damage (*Li et al., 2020*; *Qiu et al., 2021*). Thus, we hypothesized that aromatase defects might lead to a decrease in estradiol levels and an increase in androstenedione, causing an inflammatory response and leading to retinal ganglion cell death in patient with glaucoma (*Figure 10B*). In addition, androstenedione converted to estrogens is catalyzed by the aromatase (*Figure 10C*). Furthermore, some of the metabolites might also be linked to PACG physiological processes that remain unclear, and future research in this field is needed. Sex hormones, including androstenedione, might be associated with glaucoma types such as POAG, normal-tension glaucoma, and pseudoexfoliation glaucoma (*Bailey et al., 2018*). The precise molecular mechanisms remain unclear. More research is essential in understanding their relationship with different glaucoma types.

Furthermore, other studies have also found significant correlations between androstenedione and other diseases. For example, *García-Sánchez et al., 2022*, found that androstenedione can predict the progression of frailty syndrome in patients with localized breast cancer treated with aromatase inhibitors. *Adriaansen et al., 2022*, reported that diurnal salivary androstenedione levels in healthy volunteers can be used for monitoring the treatment efficacy in patients with congenital adrenal hyperplasia.

As far as our understanding goes, this is the initial investigation to methodically outline blood metabolites and scrutinize the diagnostic efficacy of a potential biomarker for PACG. Nevertheless, our study is not without its limitations: (1) The supplementary phase of our study was conducted with a restricted number of patients and healthy controls. Therefore, further investigations with larger sample sizes are necessary to confirm the diagnostic accuracy of serum androstenedione for the detection of PACG. (2) Our study cohort exhibited uniform genetic and environmental traits, which may restrict the generalizability of our findings to populations with diverse ethnic or racial backgrounds. (3) Despite the matching of PACG patients and controls for age, gender, BMI, hypercholesterolemia, hypertension, diabetes, smoking, and drinking, it is possible that unidentified residual confounding factors could impact the observed metabolomic disparities. (4) Throughout the duration of the follow-up period, the majority of individuals received pharmacological treatment, and the study's follow-up period was limited to a mere 2 years. Consequently, the outcomes may have been impacted by either behavioral adjustments prompted by patients' cognizance of their medical condition or by any form of therapeutic intervention. (5) Despite our examination of the impact of hormone intake (including estrogen, progestagen, and anti-androgen), the potential influence of reproductive aging cannot be entirely dismissed. (6) Understanding the link between changes in androstendione levels and glaucoma severity might hinge on metabolic and anti-inflammatory pathways. However, the current study did not delve deeply into the mechanism verification. Further research is essential to comprehensively grasp the precise relationship between androstendione and the severity of glaucoma, as well as the mechanisms at play. In conclusion, androstenedione has been identified and validated through the use of widely targeted LC-MS or targeted chemiluminescence immunoassay, demonstrating its ability to effectively differentiate between healthy individuals and those with PACG. Additionally, baseline androstenedione levels may serve as a valuable predictor of glaucomatous VF progression. However, further clinical and basic studies are required to confirm the clinical utility of serum androstenedione for early-stage PACG diagnosis, as well as its potential for monitoring/predicting VF progression and elucidating the underlying mechanisms linking androstenedione to PACG.

## Acknowledgements

The authors also thank Wuhan Metware Biotechnology Co., Ltd. for assistance of mass spectrometric analysis. The funder of the study had no role in study design, data collection, data analysis, data interpretation, or writing of the report.

## Additional information

### Funding

| Funder | Grant reference number | Author |
| --- | --- | --- |
| National Natural Science Foundation of China | 82302582 | Shengjie Li |
| Youth Medical Talents – Clinical Laboratory Practitioner Program | 2022-65 | Shengjie Li |
| Shanghai Municipal Health Commission Project | 20224Y0317 | Shengjie Li |
| Higher Education Industry-Academic-Research Innovation Fund of China | 2023JQ006 | Wenjun Cao |

The funders had no role in study design, data collection and interpretation, or the decision to submit the work for publication.

### Author contributions

Shengjie Li, Conceptualization, Data curation, Software, Formal analysis, Funding acquisition, Validation, Investigation, Methodology, Writing – original draft, Writing – review and editing; Jun Ren, Data curation, Software, Validation, Investigation, Methodology, Writing – review and editing; Zhendong Jiang, Data curation, Software, Investigation, Methodology, Writing – review and editing; Yichao Qiu, Data curation, Validation, Methodology, Writing – review and editing; Mingxi Shao, Resources, Investigation, Methodology, Writing – review and editing; Yingzhu Li, Data curation, Investigation, Writing – review and editing; Jianing Wu, Software, Formal analysis, Methodology, Writing – review and editing; Yunxiao Song, Resources, Data curation, Methodology, Writing – review and editing; Xinghuai Sun, Conceptualization, Resources, Funding acquisition, Project administration, Writing – review and editing; Shunxiang Gao, Resources, Data curation, Software, Formal analysis, Validation, Methodology, Project administration, Writing – review and editing; Wenjun Cao, Conceptualization, Data curation, Software, Formal analysis, Funding acquisition, Validation, Investigation, Methodology, Project administration, Writing – review and editing

### Author ORCIDs

Shengjie Li (iD) https://orcid.org/0000-0002-6443-740X
Shunxiang Gao (iD) https://orcid.org/0000-0001-5267-6289

### Ethics

This study was approved by the Ethics Committee of Eye and ENT Hospital of Fudan University (2020[2020013]) and was conducted under the Declaration of Helsinki. All participants provided written informed consent prior to their participation.

Reviewer #1 (Public Review): https://doi.org/10.7554/eLife.91407.3.sa1
Reviewer #2 (Public Review): https://doi.org/10.7554/eLife.91407.3.sa2
Author Response https://doi.org/10.7554/eLife.91407.3.sa3

## Additional files

### Supplementary files

• Supplementary file 1. List of statistical approach and packages.

• Supplementary file 2. The clinical and demographic characteristics of primary angle closure glaucoma (PACG) and cataract subjects in the supplemental phase.

• Supplementary file 3. The clinical and demographic characteristics of same primary angle closure glaucoma (PACG) patients between pre-treatment and post-treatment.

• Supplementary file 4. The differential metabolites associated with primary angle closure glaucoma

(PACG) and their fold changes in discovery set 1.

• Supplementary file 5. The differential metabolites associated with primary angle closure glaucoma (PACG) and their fold changes in discovery set 2.

• Supplementary file 6. The detecting parameters of 32 differential metabolites.

• Supplementary file 7. The relationship between androstenedione and risk of primary angle closure glaucoma (PACG).

• Supplementary file 8. The results of receiver operating characteristic (ROC) curve.

• Supplementary file 9. Comparison of characteristics of no progression and progression group in primary angle closure glaucoma (PACG) patients.

• Supplementary file 10. Comparison of areas under the receiver operating characteristic curve (AUCs) value among docosahexaenoic acid (DHA), free fatty acid (FFA) (22:6), free fatty acid (FFA) (18:4), and androstenedione.

• MDAR checklist

## Data availability

The mass spectrometry proteomics data have been deposited to the ProteomeXchange Consortium via the iProX partner repository with the dataset identifier PXD048826.

The following dataset was generated:

| Author(s) | Year | Dataset title | Dataset URL | Database and Identifier |
|---|---|---|---|---|
| Li S, Ren J, Jiang Z, Qiu Y, Shao M, Li Y, Wu J, Song Y, Sun X, Gao S, Cao W | 2024 | Serum Metabolomics in Patients with Ocular Diseases | https://proteomecentral.proteomexchange.org/cgi/GetDataset?ID=PXD048826 | ProteomeXchange, PXD048826 |

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
