## [Editor Report · eLife assessment]

This study presents a **valuable** finding that serum androstenedione levels may provide a new biomarker for early detection and progression of glaucoma, although a single biomarker is unlikely to be singularly predictive due to the etiological heterogeneity of the disease. The strength of the evidence presented is **solid**, supported by multiple lines of evidence.

---

## [Referee Report · Reviewer #1 (Public Review)]

Glaucoma is the leading cause of irreversible blindness worldwide, affecting more than 80 million people. Primary open angle glaucoma (POAG) is the prevalent form of glaucoma, while prevalence of primary angle closure glaucoma (PACG) is highest in Asia compared to over the world. Early detection of glaucoma and severity prediction is mandatory, and therefore the main aim of this study focused on characterizing the metabolite profile associated with PACG, identify potential blood diagnostic markers, assess their specificity for PACG and verify their applicability to predict progression of visual field loss. To this end, Li et al. implemented a 5-phases multicenter prospective study to identify novel candidate biomarkers of PACG. A total of 616 individuals were recruited, identifying 1464 distinct metabolites in the serum by metabolomics and chemiluminescence immunoassays. By applying different machine learning algorithms the metabolite androstenedione showed good discrimination between PACG and control subjects, both the discovery and validation phases. This metabolite also showed alterations in the aqueous humor and higher levels of androstenedione seemed to be associated with faster loss of visual field. Overall, the authors claimed that serum androstenedione levels may provide a new biomarker for early detection and monitoring/predicting PACG severity/progression.

Strengths:

• Omics research on glaucoma is constrained by inadequate sample sizes, a dearth of validation sets to corroborate findings and absence of specificity analyses. The 5-phases study designed try overcoming these limitations. The proposed study design is very robust, with well described discovery set (1 and 2), validation phase (1 and 2), supplemental phase and cohort phase. Large and well-characterized patients with adequate control subjects contributed to the robustness of the results.

• Combining untargeted and targeted metabolomics using mass spectrometry instruments (high resolution and low resolution) with an additional chemiluminiscence immunoassay determining androstenedione levels

• Androstenedione achieved better diagnostic accuracy across the discovery and validation sets, with AUC varying between 0.85 and 1.0. Interestingly, baseline androstenedione levels can predict glaucoma progression via visual field loss results.

• Positive correlation was observed between levels of androstenedione in serum and aqueous humor of PACG patients.

• A level higher of 1.66 ng/mL of the metabolite androstenedione seems to imply high risk of visual field loss. Androstenedione may serve as predictor of glaucomatous visual field progression.

Weaknesses:

• A single biomarker seems very unlikely to be of much help in the detection of glaucoma due to the etiological heterogeneity of the disease, the existence of different subtypes, and the genetic variability among patients. Rather, a panel of biomarkers may provide more useful information for clinical prediction, including better sensitivity and specificity. The inclusion of additional metabolites already identifying in the study, in combination, may provide more reliable and correct assignment results.

• The number of samples in the supplementary phase is low, larger samples sizes are mandatory to confirm the diagnostic accuracy.

• Cohorts from different populations are needed to verify the applicability of this candidate biomarker.

• Sex hormones seem to be associated also with other types of glaucoma, such as primary open-angle glaucoma (POAG), although the molecular mechanisms are unclear (see doi:10.1167/iovs.17-22708). The inclusion of patients diagnosed with other subtypes of glaucoma, like POAG, may contribute to determine the sensitivity and specificity of the proposed biomarker. Androstenedione levels should be determined in POAG, NTG or PEXG patients.

• In addition, the levels of androstenedione were found significantly altered during other diseases as described by the authors or by conditions like polycystic ovary syndrome, limiting the utility of the proposed biomarker.

• Uncertainty of the androstenedione levels compromises its usefulness in clinical practice.

---

## [Referee Report · Reviewer #2 (Public Review)]

Summary:

The objective of authors using metabolomics analysis of primary angle closure glaucoma (PACG) is to demonstrate that serum androstenedione is a novel biomarker that can be used to diagnose PACG and predict visual field progression.

Strengths:

Use of widely targeted and untargeted metabolite detection conditions. Use of liquid chromatography-tandem mass spectrometry and a chemiluminescence method for confirmation of androstenedione.

The authors have incorporated the relevant changes in their manuscript and improved the presentation.

---

## [Author Response]

The following is the authors’ response to the original reviews.

**Reviewer #1 (Public Review):**
1. A single biomarker seems very unlikely to be of much help in the detection of glaucoma due to the etiological heterogeneity of the disease, the existence of different subtypes, and the genetic variability among patients. Rather, a panel of biomarkers may provide more useful information for clinical prediction, including better sensitivity and specificity. The inclusion of additional metabolites already identifying in the study, in combination, may provide more reliable and correct assignment results.

The authors’ answer: Thank you for your comment. We recognize the constraints of using single biomarkers for diagnosis. In upcoming research, we aim to incorporate multiple biomarkers to improve diagnostic accuracy and will consider adding more metabolites as suggested.

2. The number of samples in the supplementary phase is low, larger sample sizes are mandatory to confirm the diagnostic accuracy.

The authors’ answer: Thank you for your comment. Collecting aqueous humor is invasive, making samples scarce. We acknowledge the small sample size limitation. In future studies, we plan to use larger samples to verify the biomarker's diagnostic accuracy. Your feedback emphasizes the need for thorough validation in our next research

3. Cohorts from different populations are needed to verify the applicability of this candidate biomarker.

The authors’ answer: Thank you for the suggestion. We agree on the need to test the biomarker's relevance across varied populations. Reports from other groups will help confirm and broaden our results.

4. Sex hormones seem to be associated also with other types of glaucoma, such as primary open-angle glaucoma (POAG), although the molecular mechanisms are unclear (see doi:10.1167/iovs.17-22708). The inclusion of patients diagnosed with other subtypes of glaucoma, like POAG, may contribute to determining the sensitivity and specificity of the proposed biomarker. Androstenedione levels should be determined in POAG, NTG, or PEXG patients.

The authors’ answer: I agree with your comment and thank you for your suggestion. PACG is a major cause of irreversible blindness in Asians. While this study centers on PACG, the link between sex hormones and other glaucoma subtypes, like POAG, merits investigation. Future studies will include POAG and other subtypes to further assess androstenedione's diagnostic relevance.

5. In addition, the levels of androstenedione were found significantly altered during other diseases as described by the authors or by conditions like polycystic ovary syndrome, limiting the utility of the proposed biomarker.

The authors’ answer: Thank you for your advice. Androstenedione levels also change in conditions like polycystic ovary syndrome, which could affect the biomarker's specificity. We plan to further study androstenedione's unique changes in glaucoma versus other conditions to clarify its diagnostic value.

6. Uncertainty of the androstenedione levels compromises its usefulness in clinical practice.

The authors’ answer: The uncertainty surrounding androstenedione levels and its impact on clinical applicability is a valid concern. We plan to delve deeper into understanding the variability and determinants of androstenedione levels to better assess its clinical relevance.

**Reviewer #2 (Public Review):**
The "predict" part is on much less solid ground. The visual field progression and association with serum androstenedione within the current experimental design eludes to a correlation. It truly cannot be stated as predictive. To predict one needs to put the substance when nothing is there and demonstrate that the desired endpoint is reached. Conversely, the substance (androstenedione) can be removed, and show that the condition regresses. None of these are possible without model system experiments, which have not been done. The authors could put some additional details in the methods, such as: (1) how much sample was collected, (2) whether equal serum volume for analysis had equal serum proteins (or cells). They have used a LC-MS/MS and a Chemiluminescence method, but another independent method such as GC-MS/MS or NMR to detect androstenedione for a subset of patients with different stages of visual field defect would be desirable.

The authors’ answer: We acknowledge your constructive critique concerning our use of the term "predict". In the present study, we elucidated a discernible correlation between visual field progression and serum androstenedione concentrations. We are cognizant of the critical distinction between correlation and causation, and we concur that our application of the term “predict” may have been overly assertive in this context.

Your emphasis on the imperative of employing model system experiments to unequivocally ascertain causative relationships is well-received. The experimental approach of modulating the substance, androstenedione in this case, to empirically observe its consequential impact on the condition, is a pivotal direction that warrants exploration in subsequent research endeavors.With regard to the variability of serum protein concentrations across participants, we adopted a methodological standardization by ensuring that the analyzed serum volume remained consistent across samples. This was implemented to enhance the reliability and generalizability of our findings.

Your recommendation to consider alternative detection methodologies, specifically GC-MS/MS or NMR, is duly noted. Although our choice of LC-MS/MS and Chemiluminescence was predicated on available resources, we recognize the scientific merit in leveraging multiple analytical techniques. In future investigations, we endeavor to incorporate a broader spectrum of detection methodologies for androstenedione, particularly when assessing patients with varied visual field defect stages, thereby bolstering the robustness and validity of our conclusions.

**Reviewer #1 (Recommendations for The Authors):**
1. POAG is the leading cause of irreversible blindness worldwide (see reference #4). The prevalence of PACG is highest in Asia, but the major form of glaucoma is still POAG. The authors should modify the abstract and background sections accordingly (see line 30 and lines 61-62).

The authors’ answer: Thank you for your suggestion, and we apologize for this mistake. The sentence” Primary angle closure glaucoma (PACG) is the leading cause of irreversible blindness worldwide” has been changed to” Primary angle closure glaucoma (PACG) is the leading cause of irreversible blindness in Asia”. (Page 2, lines 33; Page 3, lines 62-64)

2. Line 69, please change the sentence "the He et al. taught us..." to the following "the He et al. study taught us.".

The authors’ answer: Thank you for your comment. The sentence "the He et al. taught us..." has been changed to "the He et al. study taught us.". (Page 3, lines 72)

3. I suggest including the name of the identified candidate biomarker in the title of the manuscript. The title must be straightforward.

The authors’ answer: We agree with your comment and thank you for your suggestion. The sentence “Metabolomics Identifies and Validates Serum Novel Biomarker for Diagnosing Primary Angle Closure Glaucoma and Predicting the Visual Field Progression” has been changed to “Metabolomics Identifies and Validates Serum Androstenedione as Novel Biomarker for Diagnosing Primary Angle Closure Glaucoma and Predicting the Visual Field Progression”. (Page 1, lines 1)

4. Line 88, please change "normal subjects" to "control individuals".

The authors’ answer: Thank you for your comment. We have changed "normal subjects" to "control individuals”. (Page 4, lines 91)

5. Line 95 and so on along the manuscript, avoid the term "normal controls" or "normal" and use only the term "controls".

The authors’ answer: Thank you for your advice. "normal subjects" has been changed to "controls". (Page 4, lines 113; Page5, lines 118,120,124,128,133)

6. In the participants section, indicate the ocular treatments of PACG patients. For example, on line 141, which "treatment" are you referring to?

The authors’ answer: Thank you for your comment. We apologize to this vague statement. Treatment included medical treatment and surgical treatment. We have revised it in the manuscript. (Page 5, lines 142)

7. The entire section 2.4 is confusing. According to Figure S2, untargeted metabolomics was conducted with a mixed sample containing "all" serum extracts in order to obtain an in-house database with molecular features present in serum by LCHRMS. Then, this database was used for targeted metabolomics in individual serum samples using LCQQQ. However, as it is described in the manuscripts, it seems that first, an untargeted metabolomics analysis was carried out to identify altered metabolites, then targeted metabolomics was carried out to validate the untargeted analysis and finally, a profiling analysis was carried out to construct the database. The workflow must be clearly discussed and amended to be understable.

The authors’ answer: Thank you for your comment. We have revised the description of the experimental method section 2.4. (Page 7, lines 195-198)

8. Please, briefly explain what widely-targeted metabolomics is and how it works in this study (see section 2.4).

The authors’ answer: Thank you for your comment. For extensively targeted metabolome detection, a local database was first established by using the standard database, and ion pair information was obtained by scanning ion pairs of mixed samples (QC) with QTOF. A wide range of metabolites were qualitatively obtained by comparing with the local self-built database, and then the metabolites of each sample were qualitatively and quantitatively measured by MRM scanning mode of triple four-bar QQQ. This project combines the non-target public database scanning construction database and the wide target local database to build a new database, and then scans the database of the samples of this project with Q-TOF, and then carries out the qualitative and quantitative detection of metabolites of each sample in MRM mode. (Figure S2)

9. On Table 1, indicate the number of patients and controls with cataracts.

The authors’ answer: For the glaucoma group and the control group, we have excluded people with cataracts. This section is described in the inclusion and exclusion criteria for supplementary materials. (Inclusion and exclusion criteria)

10. On "Sample processing" section, lines 152 and 153: Have you used cold methanol to ensure metabolic quenching? If not, how metabolite quenching was carried out?

The authors’ answer: Thank you for your comment. We use cold methanol to extract metabolites, and the early blood samples have been stored in a -80°C refrigerator to ensure a low temperature process and ensure metabolic quenching. (Page 6, lines 196)

11. On the same "Sample processing" section, have you used internal standards during metabolite extraction? If yes, ones? If not, why?

The authors’ answer: Thank you for your comment. In the metabolite extraction process of each sample, the same internal standard was added, and the same volume of 50 μL serum samples were extracted. The specific internal label name has been added in "Sample processing" section. (Page 6, lines 153-155)

12. Lines 161-163, I suggest including in the supplementary material the worklist of the entire experiment run by LC-MS, including analytical replicates and QCs.

The authors’ answer: Thank you for your comment. Worklist for mass spectrometry can be found in supplementary sheet1. (Page 6, lines 165)

13. The title of the section "Detection method" does not seem appropriate, please change it to "Analytical methods "or something similar.

The authors’ answer: Thank you for your advice. "Detection method" has been changed to “Analytical methods “. (Page 6, lines 168)

14. Section 2.4.1, I suggest changing "Untargeted detection conditions" to "Untargeted metabolomics analysis".

The authors’ answer: Thank you for your comment. "Untargeted detection conditions" has been changed to "Untargeted metabolomics analysis". (Page 6, lines 169)

15. Lines 170-172, the column used is compatible with 100% water, why start with 5% acetonitrile?

The authors’ answer: Thank you for your comment. If the acetonitrile starting gradient is 0, it will cause a lot of water-soluble substances to elute and easily clog the column, so we want to use 5% organic phase.

16. Section 2.4.1, the chromatographic conditions (mobiles phases) were the same in both positive and negative ion mode? It is desirable to change or adjust a basic pH when working in negative, so please amend and clarify it.

The authors’ answer: Thank you for your comment. In the negative ion mode, the peak shape of the chromatogram under the acidic system is better than that under the alkaline system, so we choose the acidic system.

17. I am not able to clearly understand what is "widely targeted conditions" (see section 2.4.2). What is the difference with the conventional targeted metabolomics analysis? In my view, widely-targeted metabolomics refers to the combination of untargeted metabolomics and targeted metabolomics. This must be clarified and simplified.

The authors’ answer: Thank you for your syggestion. The characterization of metabolites in this study was conducted using a non-targeted database and a self-built database. Non-targeted metabolites were characterized with mixed samples, and then combined with the laboratory self-established database to form a new metabolome database for this study. 2.4.2 The broad targeting here refers to the use of the MWDB standard self-built database to characterize metabolites, and then the QQQ MRM model to quantify metabolites. In order to clearly describe the detection process, this part of the method has been modified. (Figure S2)

18. Line 199, please, indicate the normalization carried out.

The authors’ answer: We agree with your comment and thank you for your suggestion. The normalization description is missing from its data processing steps and has been corrected in the manuscript. (Page 7, lines 203)

19. How many instrumental replicates have you carried out both in untargeted and targeted metabolomics? Please, indicate it.

The authors’ answer: Thank you for your advice. In this project, all sample mixtures were used as QC samples, which were repeated several times in the testing process (one QC sample was inserted between every 10 samples), and the repeated correlation between repeated QC was more than 99% to ensure the stability of sample testing. (Sheet1)

20. Line 267, why did you select a fold changes threshold greater than 1.15 (or lower 0.85)? In metabolomics, it would be desirable to have a minimum of 1.5-fold change considering the variability of data.

The authors’ answer: Thank you for your comment. FC reduction is selected to expand potential candidate metabolites and can be repeated in three batches and refer to the literature "Blood metabolomics uncovers inflammation-associated mitochondrial dysfunction as a potential mechanism. underlying ACLF "method screening threshold.

21. To include anywhere the molecular formula of androstenedione.

The authors’ answer: I agree with your comment and thank you for your suggestion. We have added the molecular formula of androstenedione to the supplementary material. (Page 17, lines 475)

22. Line 290 is not Figure 4B and 4C, you may refer to Figure 3B and 3C.

The authors’ answer: Thank you for your advice. We apologize to this mistake. Figure 4B and 4C have been changed to Figure 3B and 3C.

23. Figure S3 was lost from Supplementary material, please include it.

The authors’ answer: Thank you for your comment. We apologize to this mistake. There is an error in the ordering of the supplementary graph. Figure 3 is redundant, and we have modified it in the supplementary materials.

24. Figure 4 B, indicate in the text the average and uncertainty of androstenedione levels in both control and PACG groups.

The authors’ answer: Thank you for your comment. In the manuscript, We have added descriptions of mean ± standard deviation of androstendione levels in the control group and the disease group. (Page 11, lines 311-312)

25. Section 3.6. please include the average and uncertainty of androstenedione levels in males and females in both control and PACG groups.

The authors’ answer: Thank you for your advice. For 3.6 section, we supplemented the mean ± standard deviation of androstenedione levels in the control and disease groups. (Page 13, lines 350-356)

26. Figure S9 seems missing.

The authors’ answer: Thank you for your comment. We apologize to this mistake. Figures S9 has been added in the Supplementary material.

27. Lines 345-346, indicate the levels obtained for the metabolite in the compared groups.

The authors’ answer: Thank you for your suggestion. The levels of androstenedione in each group are seen in “The results from both discovery set 1 (Figure S9A, Mild:32600±17011, Moderate:33215±17855, Severe:46060±21789) and discovery set 2 (Figure S9B, Mild:27866±19873, Moderate:27057±13166, Severe:43972±19234) indicated that the mean serum androstenedione levels were significantly higher in the severe PACG group compared to the moderate and mild PACG groups (P<0.001). These findings were further validated in both validation phase 1 (Figure S9C, Mild:75726±45719, Moderate:65798±30610, Severe:94348±30858) and validation phase 2 (Figure S9D, Mild:1.121±0.3143 ng/ml, Moderate:1.461±0.4391 ng/ml, Severe:2.147±0.6476 ng/ml).” and “Notably, the level of androstenedione was found to be significantly higher in PACG patients than in normal subjects in both discovery set 1 (Figure 4B, P=0.0081, Normal:33987±11113, PACG:42852±20767) and discovery set 2 (Figure 4C, P=0.0078, Normal:31559±10975, PACG:37934±18529).”

28. Line 368, you don't need to indicate the PACG abbreviation again.

The authors’ answer: Thank you for your comment. We apologize to this mistake. I have changed " patients with PACG " to "patients". (Page 13, lines 377)

29. Figure 6, panels A and B are not labeled (i.e., commented) in the body text of the manuscript.

The authors’ answer: Thank you for your suggestion. We’re very sorry for this mistake. Figure 6, panels A and B have been labeled in the manuscript. (Page 13, lines 377-379)

30. Section 3.7., when you indicate "after therapy" are you referring to surgical treatment? Please, clarify.

The authors’ answer: Thank you for your comment. We apologize to this vague statement. Blood samples were taken before and three months after surgery. “therapy” has been changed to “surgical treatment” in the manuscript. (Page 13, lines 377)

31. Line 370, "97th patient" should be replaced by "nine patients"?

The authors’ answer: Thank you for your advice. We apologize to this mistake. "97th patient" has been changed to “nine patients". (Page 13, lines 378-379)

32. Lines 370-372, it difficult to understand, please clarify why these findings indicate that severity is related to increased PACG according to Figure 6B.

The authors’ answer: Thank you for your comment. We’re very sorry for this vague statement. The sentence of “These findings showed that the levels of androstenedione that were tightly connected with PACG severity rose dramatically as PACG progressed.” Has been removed.

33. Line 447, the word "corrected" should be changed to "correlated"?

The authors’ answer: Thank you for your comment. "corrected" has been changed to "correlated". (Page 16, lines 453,456)

34. According to the literature, the levels found in control subjects are within the range of the "normal" values, i.e., are they comparable?

The authors’ answer: Thank you for your advice. Androstenedione ranges from 0.4 to 2 in the normal population. The mean standard deviation of androstenedione in the normal population was 1.552 ± 0.4859.

35. Lines 471-474, why "steroid hormone biosynthesis appears to be the critical node to high-match PACG pathophysiological concepts" while the high enrichment was observed in the "metabolic pathways"?

The authors’ answer: Metabolic pathways encompass a series of chemical reactions within a cell that enable the synthesis or breakdown of molecules to maintain the cell's energy balance. Steroid hormone biosynthesis is one of these metabolic pathways, and its products, steroid hormones, participate in a wide range of physiological processes, including metabolism, immune response, and the regulation of inflammation. In a different context, a study related to fatigue during Androgen Deprivation Therapy (ADT) showed a significant difference in metabolite levels within the steroid hormone biosynthesis pathways, emphasizing the role these pathways play in metabolic alterations. The mentioned findings suggest that steroid hormone biosynthesis and metabolic pathways are intertwined. (Page 17, lines 481-488)

36. Figure S13 and Figure S14A are the same.

The authors’ answer: Thank you for your comment. Figure S14A has been removed.

37. On lines 476-485, it would be interesting to discuss whether alterations of this metabolite could be a cause or consequence of PACG.

The authors’ answer: Based on the literature found, androstenedione is a naturally occurring steroid hormone produced by the gonads and adrenal glands, and serves as an intermediate in testosterone biosynthesis (Androstenedione (a Natural Steroid and a Drug Supplement): A Comprehensive Review of Its Consumption, Metabolism, Health Effects, and Toxicity with Sex Differences). Early events in the pathobiology of glaucoma involve oxidative, metabolic, or mechanical stress acting on retinal ganglion cells (RGCs), leading to their rapid release of danger signals such as extracellular ATP, thus triggering microglial and macroglial activation as well as neuroinflammation (Immune Responses in the Glaucomatous Retina: Regulation and Dynamics). However, one might speculate that since androstenedione is a steroid hormone, it could potentially impact the inflammatory and metabolic stress observed in the pathophysiological processes of glaucoma (Adaptive responses to neurodegenerative stress in glaucoma). Metabolic and anti-inflammatory avenues might be crucial in understanding the relationship between alterations in androstenedione levels and the severity of glaucoma. Nevertheless, more research and literature analysis would be necessary to better understand the precise relationship and its underlying mechanisms between these two entities.

38. I suggest sending the MS and MS/MS into a publicly available repository.

The authors’ answer: Thank you for your suggestion. Further research will necessitate the utilization of the raw mass spectrometry data. We anticipate making this raw data available in a public repository upon the conclusion of subsequent experiments.

**Reviewer #2 (Recommendations for The Authors):**
The authors should aim to describe methods in greater detail.The authors could improve the writing to accurately describe their results and their interpretation and state what else could be done to make the result truly "predictive".

The authors’ answer: (1) Detail Enhancement in the Methods section: We expand the description of methods such as sample pre-processing, mass spectrometry detection, and result analysis in the study to provide more detailed information about the procedures, equipment, and materials used. (2) Improvement in Writing Quality: We have engaged a scientific editor to review our manuscript for clarity, coherence, and consistency to ensure that the results and interpretations are accurately and clearly conveyed. Terminologies and phrases have been revised to better reflect the findings and interpretations. (3) Limitation supplement: We have included a discussion on the limitations of our study and suggested additional studies and analyses that could be conducted to enhance the predictive value of our findings. We sincerely appreciate the constructive feedback from the reviewer, which has greatly contributed to improving the quality and rigor of our manuscript.